# Exploring the Effectiveness of Object-Centric Representations in Visual Question Answering: Comparative Insights with Foundation Models

**Amir Mohammad Karimi Mamaghan**[1]    **Samuele Papa**[2]    **Karl Henrik Johansson**[1,3]
**Stefan Bauer**[4,5]    **Andrea Dittadi**[4,5,6]

[1]KTH Royal Institute of Technology    [2]University of Amsterdam    [3]Digital Futures
[4]Helmholtz AI    [5]Technical University of Munich    [6]MPI for Intelligent Systems, Tübingen

## Abstract

Object-centric (OC) representations, which model visual scenes as compositions of discrete objects, have the potential to be used in various downstream tasks to achieve systematic compositional generalization and facilitate reasoning. However, these claims have yet to be thoroughly validated empirically. Recently, foundation models have demonstrated unparalleled capabilities across diverse domains, from language to computer vision, positioning them as a potential cornerstone of future research for a wide range of computational tasks. In this paper, we conduct an extensive empirical study on representation learning for downstream Visual Question Answering (VQA), which requires an accurate compositional understanding of the scene. We thoroughly investigate the benefits and trade-offs of OC models and alternative approaches including large pre-trained foundation models on both synthetic and real-world data, ultimately identifying a promising path to leverage the strengths of both paradigms. The extensiveness of our study, encompassing over 600 downstream VQA models and 15 different types of upstream representations, also provides several additional insights that we believe will be of interest to the community at large.

## 1 Introduction

Object-centric (OC) learning aims to represent the physical world's inherent structure, assuming visual scenes consist of entities or objects and employing this as an inductive bias for neural networks (Goyal et al., 2019; Locatello et al., 2020; Lin et al., 2020b; Burgess et al., 2019; Singh et al., 2022a;b; Seitzer et al., 2023; Wu et al., 2023; Jiang et al., 2023; Löwe et al., 2023). Applied in various domains like visual reasoning (Chen et al., 2021b; Ding et al., 2021a; Wu et al., 2022; Ding et al., 2021b; Santoro et al., 2017; Webb et al., 2023; Mondal et al., 2023) and image and video generation (Chen et al., 2021a; Singh et al., 2022a;b; Elsayed et al., 2022; Jabri et al., 2023), these representations play a crucial role in capturing compositional and causal structures, with the potential to improve the generalizability and interpretability of AI algorithms (Lake et al., 2017; Schölkopf et al., 2021; Brady et al., 2023; Kim et al., 2023b; Jung et al., 2024). Breaking down scenes into conceptual elements corresponding to causal factors aligns with the idea that causal models play a crucial role in achieving human-level generalization (Pearl, 2009; Peters et al., 2017; Mansouri et al., 2023; Liu et al., 2023).

While OC representations thus provide a way of representing the state of a visual scene, a comprehensive understanding of these representations is still an ongoing exploration. Recently, there have been several works on evaluating OC representations. Some studies evaluate object-centric models in terms of reconstruction and segmentation accuracy, and quantify the quality and information content of object representations via a downstream object property prediction task (Dittadi et al., 2022; Papa et al., 2022). Arguing that a major goal of representation learning is to facilitate downstream tasks, Yoon et al. (2023) focuses on the evaluation of the representations in reinforcement learning, which requires a thorough understanding of the environment in terms of objects and the relations between them. However, a more direct quantification of the role of object-centric representations for reasoning is still missing.

In the rapidly evolving landscape of deep learning, foundation models, often characterized by self-supervised and large-scale pre-training, have demonstrated unparalleled capabilities in generalization and zero-shot learning, showcasing their prowess in tasks across diverse domains from natural language processing to computer vision (Kirillov et al., 2023; Wen et al., 2023; Touvron et al., 2023; Brown et al., 2020; Chowdhery et al., 2023; Huang et al., 2023; Borsos et al., 2023; Yang et al., 2023; Jumper et al., 2021; Rombach et al., 2022). Despite their widespread success, foundation models have not been comprehensively analyzed and compared with OC models.

In this paper, we aim to take one step towards understanding the relevance of object-centric representations in the era of foundation models by evaluating them on visual reasoning tasks through a simple framework. More specifically, **our main contributions** are the following:

- We conduct a large empirical study on representation learning for downstream Visual Question Answering (VQA) (Antol et al., 2015; Johnson et al., 2017) on three synthetic and two real-world multi-object datasets. In our extensive evaluation, we train overall 684 downstream transformer models for VQA, involving 15 different types of upstream representation models, ranging from VAEs to state-of-the-art OC methods to large pre-trained foundation models.

- We identify and investigate the trade-offs between large foundation models and OC models. We observe that, without any fine-tuning or hyperparameter adjustment, foundation models perform comparably to the top-performing OC models. On the other hand, they typically require more compute and larger downstream models. We find that applying the OC inductive bias to foundation models effectively achieves the best of both worlds, reducing the downstream computational needs while achieving comparable or better performance and obtaining more explicit representations.

- We present several additional insights regarding, among other things, the correlation between performances on VQA and a simpler downstream task, the relationship between upstream and downstream performance of OC models, the effect of training set size on VQA performance, the difference between different question types, and a deeper analysis of the global (single-vector) representations of traditional VAEs.

## 2 RELATED WORKS

**Object-Centric Learning.** Object-centric (OC) learning has gained attention over the past few years (Goyal et al., 2019; Singh et al., 2021; 2022a;b; Seitzer et al., 2023; Wu et al., 2021; 2023; Jiang et al., 2023; Eslami et al., 2016; Crawford & Pineau, 2019; Kosiorek et al., 2018; Jiang et al., 2019; Dittadi & Winther, 2019; Engelcke et al., 2019; 2021; Lin et al., 2020a;b; Greff et al., 2017; Gregor et al., 2015; Yuan et al., 2019; Locatello et al., 2020; Burgess et al., 2019; Jabri et al., 2023; Chen et al., 2021a; Kipf et al., 2019; 2022; Elsayed et al., 2022; Löwe et al., 2023; Kori et al., 2023; Sajjadi et al., 2022; Daniel & Tamar, 2022). OC models aim to learn visual representations without supervision by treating each image as a composition of objects. Among them, Slot Attention (Locatello et al., 2020) stands out as a popular model and a crucial component in several recent state-of-the-art models. Numerous enhancements have been proposed, including improvements of the Slot Attention module (Biza et al., 2023; Jia et al., 2023; Majellaro & Collu, 2024) or adding additional modules on top (Kim et al., 2023a), using a transformer decoder instead of the original mixture-based decoder (Singh et al., 2021; 2022b), replacing the CNN backbone with a pre-trained model (Seitzer et al., 2023), and integrating diffusion models with Slot Attention (Jiang et al., 2023; Wu et al., 2023; Jabri et al., 2023).

**Evaluation of Object-Centric Representations.** OC methods have been applied in several works in visual reasoning (Chen et al., 2021b; Ding et al., 2021a; Wu et al., 2022; Ding et al., 2021b; Santoro et al., 2017; Webb et al., 2023; Mondal et al., 2023; Driess et al., 2023) and some of these works try to address the Visual Question Answering task itself. Ding et al. (2021a) propose a new method to address the VQA in videos and run a transformer over slots obtained from a pre-trained MONet (Burgess et al., 2019), and text tokens of the question, and applies an MLP on top to predict the answer. The method proposed by Wu et al. (2022) reasons over the object representations of Slot Attention to model spatiotemporal relationships, and predicts future object states. Their framework is also applied to a VQA downstream task.

In addition, a few works focus more specifically on the evaluation of OC representations. Weis et al. (2021) designs a benchmark over only OC video models and analyzes their performance over different tracking scenarios relevant to natural videos. Yang & Yang (2024) evaluates OC representations and shows their shortcomings in segmenting objects in a real-world dataset. Dittadi et al. (2022) evaluates

the representations indirectly in the context of reconstruction loss, segmentation quality, and object property prediction, and analyzes their generalization and robustness. Papa et al. (2022) uses the same evaluation metrics on a dataset with complex textures. Yoon et al. (2023) evaluates the representations on more practically relevant downstream tasks in reinforcement learning and includes a wider range of methods compared to the previous works. Finally, Driess et al. (2023) demonstrates the suitability of OC representations in planning and VQA tasks within a robotic environment. However, the assessment is done on a single OC baseline in the presence of a Large Language Model (LLM) and the VQA setup is restricted to particular scenarios. In our work, we are interested in investigating the suitability of different types of representation, including object-centric ones, for reasoning tasks. To this end, we opt to more directly assess the suitability of representations for reasoning through VQA.

# 3 EXPERIMENTAL SETUP

In this section, we provide an overview of our experimental setup. First, we introduce the downstream task used in our experiments to evaluate representations. We then outline the upstream representation models, the datasets and metrics, and the concrete setup for learning the downstream task.

## 3.1 VISUAL QUESTION ANSWERING

In this paper, we evaluate the performance attainable on a Visual Question Answering (VQA) task (Antol et al., 2015) from different representations of the visual scenes. With questions that can involve any number of objects from just one to all the objects in an image, VQA presents a more demanding challenge compared to object-level tasks. It requires a thorough understanding of the image and complex reasoning about objects and their relationships. We therefore choose VQA as a benchmark to directly assess the suitability of different representations for reasoning.

Given an image, the task is to answer a natural language question such as *"How many tiny green objects are made of the same material as the purple cube?"*. The questions are usually about how many objects there are, whether an object with a specific attribute exists, and what properties they have in relation to another set of objects in an image. The possible answers include "yes", "no", and various numerical and categorical values. Further details are provided in Appendix B.2.

Our framework, summarized in Fig. 1, consists of: (1) an upstream model that provides high-level representations of an image, (2) a fixed pre-trained text embedding model that converts a question in natural language to text embeddings, and (3) a downstream model that takes as input the image representation and the text embedding and outputs the answer to the question. We will elaborate on each part in the following sections.

## 3.2 UPSTREAM MODELS

To investigate OC representations, we consider three types of representations: global, fixed-region, and object-centric. Global representations encode the image into a single vector which contains high-level information about the image. Fixed-region representations consist of a fixed number of vectors, each loosely corresponding to a specific region within the image. OC representations consist of a set of vectors, each ideally corresponding to a single object.

The evaluated models are summarized in Table 1. As OC baselines, we use *MONet* (Burgess et al., 2019), *SPACE* (Lin et al., 2020b), and *Slot Attention* (*SA*) (Locatello et al., 2020). We also include *ResNet SA* (Biza et al., 2023), an improved version of the standard SA autoencoder with the following

modifications: the backbone is replaced by a ResNet34 (He et al., 2016) without pre-training; a larger feature map resolution is used in both the encoder and the decoder; and the slot initializations are learnable. We also consider *STEVE* (Singh et al., 2022b), a state-of-the-art OC video model for complex and naturalistic videos. STEVE is a more robust version of *SLATE* (Singh et al., 2022a) combining the SLATE decoder with a standard slot-level recurrent model.

Table 1: Summary of models included in our study.

| Model | Representation Type | Training Regime |
|---|---|---|
| DINOv2 (Oquab et al., 2023) | Fixed-Region | Pre-training |
| MAE (He et al., 2022) | Fixed-Region | Pre-training |
| CLIP (Radford et al., 2021) | Fixed-Region | Pre-training |
| VQ-AE (Rombach et al., 2022) | Fixed-Region | Pre-training |
| KL-AE (Rombach et al., 2022) | Fixed-Region | Pre-training |
| ResNet50 (He et al., 2016) | Fixed-Region | Pre-training |
| CNN (Zambaldi et al., 2018) | Fixed-Region | End-to-End Training |
| MultiCNN (Kipf et al., 2019) | Object-Centric | End-to-End Training |
| Slot Attention (Locatello et al., 2020) | Object-Centric | Dataset-Specific Pre-training |
| ResNet Slot Attention (Biza et al., 2023) | Object-Centric | Dataset-Specific Pre-training |
| MONet (Burgess et al., 2019) | Object-Centric | Dataset-Specific Pre-training |
| SPACE (Lin et al., 2020b) | Object-Centric | Dataset-Specific Pre-training |
| STEVE (Singh et al., 2022b) | Object-Centric | Dataset-Specific Pre-training |
| DINOSAURv2 (Seitzer et al., 2023) | Object-Centric | Dataset-Specific Pre-training |
| VAE (Watters et al., 2019) | Global | Dataset-Specific Pre-training |

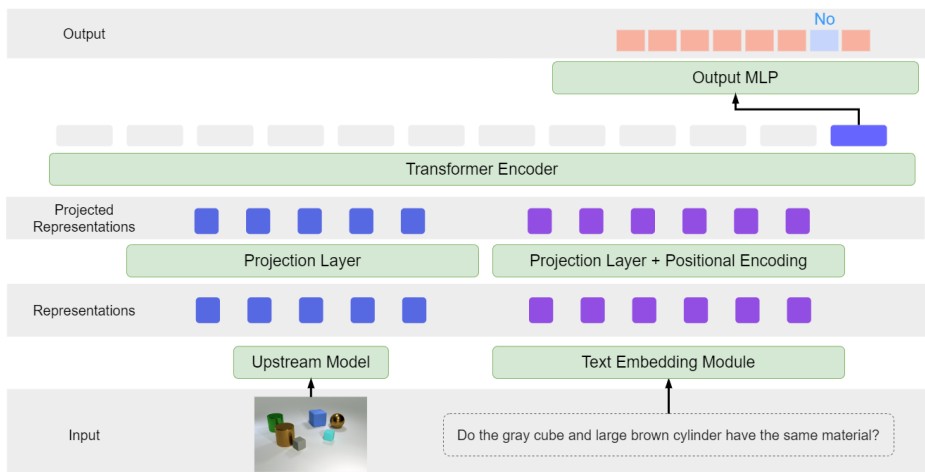

Figure 1: An overview of our framework. Starting from an image and a question, we first extract image and question representations, by applying the upstream model and the text embedding module, respectively. The obtained representations are then passed to the projection layer and then, a positional encoding is applied to the text representations. Next, both are concatenated and a transformer model is applied to the combined sequence. Finally, The answer to the question is obtained by an MLP that takes the transformed value of the *CLS* token and produces a probability vector over all possible answers.

To adapt STEVE to images, we simply consider images as 1-frame videos, following the authors' recommendation. Furthermore, as the last OC baseline, we consider *DINOSAUR* (Seitzer et al., 2023), a state-of-the-art OC image model, and replace its pre-trained DINO (Caron et al., 2021) backbone with DINOv2 (Oquab et al., 2023)—we refer to this model as *DINOSAURv2*. Following previous work, we also consider multiple CNNs (Kipf et al., 2019; Watters et al., 2017; Yoon et al., 2023), each CNN being expected to capture one object in the image, and train all of them end-to-end together with the downstream model—we refer to this approach as *MultiCNN*.

As a classic benchmark for fixed-region representations (Yoon et al., 2023; Seitzer et al., 2023), we include a pre-trained *ResNet50* (He et al., 2016). We also utilize two pre-trained autoencoders from Latent Diffusion Models (LDM) (Rombach et al., 2022), one with a KL regularization and the other one with a vector quantization layer, both with a scaling factor of 16. We refer to them as *KL-AE* and *VQ-AE*, respectively. Additionally, we use pre-trained versions of *DINOv2* (Oquab et al., 2023; Darcet et al., 2023), *Masked Autoencoder* (*MAE*) (He et al., 2022), and *CLIP* (Radford et al., 2021), all of which have achieved outstanding performance as a backbone in a diverse array of tasks. Following previous works (Santoro et al., 2017; Zambaldi et al., 2018; Yoon et al., 2023), we also implement a simple CNN, and train it end-to-end with the downstream model. As a baseline providing a global representation, we follow Dittadi et al. (2022) and consider a variation of vanilla variational autoencoders (VAE) (Kingma et al., 2014; Rezende et al., 2014) with a broadcast decoder (Watters et al., 2019). Finally, to better understand the models' performances, we include, for each dataset, a baseline trained only on questions, without any information from the corresponding images.

**Training.** Regarding the training of upstream models, we have 3 different types of models: pre-trained foundation models, pre-trained dataset-specific models, and end-to-end models. Pre-trained foundation models have been trained on large-scale datasets and tasks, serving as the basis for transfer learning in various applications. DINOv2, MAE, CLIP, VQ-AE, KL-AE, and ResNet50 belong to this category that we use off-the-shelf without fine-tuning, in all experiments. Dataset-specific pre-trained models are first trained with an autoencoding objective (only using images, disregarding the questions) on the same dataset that will be used for VQA with their original training procedures and hyperparameter choices. They are subsequently frozen, similarly to foundation models. MONet, SPACE, SA, ResNet SA, STEVE, DINOSAURv2, and VAE are in this category. Finally, end-to-end models are trained from scratch alongside the downstream model to solve the VQA task directly. CNN and MultiCNN belong to this category. For more information about the upstream models, see Appendix A.1.

## 3.3 DATASETS

**Synthetic.** We utilize three popular multi-object datasets in our experiments: *Multi-dSprites* (Matthey et al., 2017), a variation of *CLEVR* (Johnson et al., 2017) with 6 objects known as *CLEVR6*

(Greff et al., 2019; Locatello et al., 2020; Dittadi et al., 2022), and *CLEVRTex* (Karazija et al., 2021) which is a variation of CLEVR featuring synthetic scenes with diverse shapes, textures and photo-mapped materials. This dataset is closer to real-world datasets in terms of visual complexity. To analyze the effect of training data size, we consider 4 different training data sizes in Multi-dSprites with 40k, 80k, 160k, and 320k unique images, with the 320k version as the default version. Each image in the multi-object datasets consists of a background with a fixed color and a set of objects with different properties. Originally, only CLEVR contains questions associated with each image. To make the other datasets applicable to the same VQA task, we augment them with several questions (roughly 40-50) for each image, by adapting the question generation mechanism of Johnson et al. (2017) to each dataset. We use this to generate different types of questions, with possible answers including "yes", "no", natural numbers up to the maximum number of objects, and all possible values of object properties. For more details about the datasets and question generation, see Appendix B.

**Real-World.** Additionally, we extend our results to real-world scenarios with the *VQA-v2* (Goyal et al., 2017; Antol et al., 2015) and *GQA* (Hudson & Manning, 2019) datasets. VQA-v2 consists of open-ended questions about images sourced from MS COCO 2014 (Lin et al., 2014), a real-world multi-object dataset. Recently, COCO has been increasingly utilized in object-centric literature (Seitzer et al., 2023; Jiang et al., 2023; Wu et al., 2023), marking a significant advancement in complexity compared to datasets typically used to evaluate object-centric models. GQA is a large-scale dataset designed for visual question answering, focusing on compositional reasoning over real-world scenes. VQA-v2 features a diverse range of possible answers. To align with the same classification pipeline used for synthetic datasets, we limit the questions to yes/no and questions with numeric answers ranging from 0 to 14. This results in a total of 17 possible answers. For more details about the dataset and the preprocessing, see Appendix B.

## 3.4 METRICS

Following previous works (Ding et al., 2021a; Wu et al., 2022), we measure performance in our VQA downstream task by average accuracy. As metrics for the upstream OC models, we use the Mean Squared Error (MSE) of the reconstructions, and 3 segmentation metrics: the Adjusted Rand Index (ARI) (Hubert & Arabie, 1985), Segmentation Covering (SC) (Arbelaez et al., 2010), and mean Segmentation Covering (mSC) (Engelcke et al., 2019). All of these metrics have been extensively used in previous studies (Locatello et al., 2020; Dittadi et al., 2022; Singh et al., 2022b; Biza et al., 2023). See Appendix C for more details about the metrics.

## 3.5 FRAMEWORK SETUP

Our VQA framework, depicted in Fig. 1, closely follows Ding et al. (2021a). Given a pair $(x, q)$ where $x \in \mathbb{R}^{3 \times H \times W}$ denotes an image of height $H$ and width $W$, and $q$ denotes a question, the task is to select the correct answer from the set of all possible answers. Since the number of answers in each dataset is relatively small, it is not necessary to generate text tokens as the answers, and similarly to Ding et al. (2021a), we stick to the simpler case of predicting a probability vector over all possible answers in the dataset. Another key aspect to consider is that our primary focus is on evaluating representations while the method for generating questions and the format of the answers hold less significance in this context.

**Image and Text Representations.** Given a data pair $(x, q)$, the upstream model computes the image representation $z$. In global representations, $z$ is a vector of size $D_{glob}$. In OC models, $z$ is a $N_{slots} \times D_{oc}$ matrix where $N_{slots}$ is the number of slots in the OC model. In fixed-region representations, $z$ is a feature map of size $P_H \times P_W \times D_{fr}$ where the first two dimensions correspond to the feature map sizes and the third dimension is the size of the representation. For more details about obtaining image representations from the upstream models, see Appendix A.1.

To embed the question $q$ from text format to word embeddings, we use the Text-to-Text Transfer Transformer (T5; Raffel et al., 2020) which outputs a matrix $t$ of size $N_{tokens} \times D_{emb}$ representing the embeddings of the tokens in the question where the dimensions correspond to the number of tokens and the embedding size, respectively. See Appendix A.2 for more details.

**Unifying Image Representations.** In order to use different types of image representations in the downstream model which follows a transformer architecture and will be explained later on, it is necessary to unify the format of representations and convert them to a sequence. We use $z$ as it is for OC representations since each slot corresponds to an object, and can be separately used as an item in

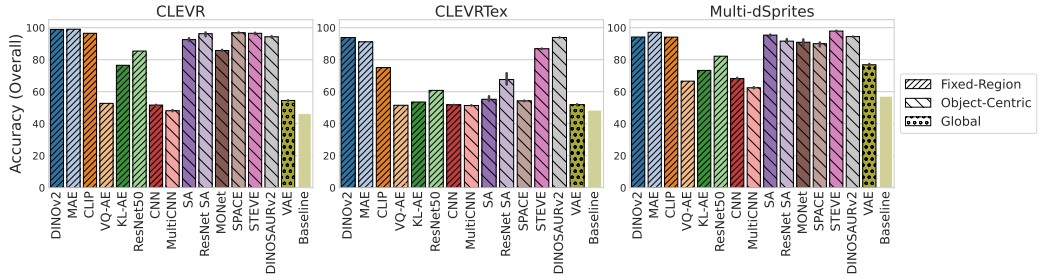

Figure 2: Average accuracies on the VQA downstream task for different upstream representation models on synthetic datasets, when using T-15 as the downstream model. The bars indicate means and 95% confidence intervals with 3 random seeds, when available.

the sequence. We reshape fixed-region representations by flattening the spatial dimensions, obtaining a matrix of size $P_H P_W \times D_{fr}$.

For global representations, we split the single vector $z$ into $K$ vectors of size $D_{glob}/K$. Here, $K$ roughly corresponds to the number of slots in an OC model. In other words, we treat $z$ as a sequence of length $K$ with a latent size of $D_{glob}/K$. While we observed this to be the most effective option in terms of downstream performance, we also considered three alternative approaches. The first applies a 2-layer MLP to $z$ and subsequently splits the output similarly to what described above; the second method treats the single vector $z$ as one token in a sequence of length 1; the third splits $z$ into $D_{glob}$ sequences of size 1. All these approaches showed poorer downstream performance, and in addition, the last one is computationally expensive due to a large sequence length.

**Downstream Model.** Following previous works on VQA (Ding et al., 2021a; Devlin et al., 2018; Lu et al., 2019), we use a transformer-based architecture (Vaswani et al., 2017). Having $t$ and the reformatted $z$ as text and image representations, we apply a separate linear layer on each to make the latent size and the embedding size equal, and we get $t'$ and $z'$, respectively. Then, to inform the downstream model about the order of words, we apply a sinusoidal positional encoding layer to $t'$. Additionally, following Ding et al. (2021a), we augment each vector in $z'$ and $t'$ with a 2-dimensional one-hot vector indicating whether the input is from the image representation or the text, and the latent size for both will become $D_{model}$. We introduce a trainable vector $CLS \in \mathbb{R}^{D_{model}}$, akin to the *CLS* token in *BERT* (Devlin et al., 2018), to generate classification results. In the final step, we concatenate $z'$, $t'$, and the *CLS* token and pass this sequence through a transformer with $N_t$ layers. An MLP classifier then takes the transformed *CLS* token and outputs a probability vector over all possible answers.

### 3.6 LIMITATIONS

While our goal is to execute a robust and informative experimental study to address the research questions identified in Section 1, it's important to acknowledge inherent limitations related to datasets, models, and evaluations. The foundation models in our study are trained with different objectives and on datasets that differ in size and characteristics, making direct comparisons with OC models more difficult. However, it is important to emphasize that this is first and foremost a pragmatic study aimed at deriving practical, actionable insights into representation learning for downstream reasoning tasks. To achieve this, we empirically investigate a diverse range of approaches directly available in the literature, without significant modifications, and evaluate their effectiveness for these tasks.

## 4 EXPERIMENTAL RESULTS

Our key findings are presented in this section. In our main set of experiments, we assess how different model representations perform on the Visual Question Answering (VQA) downstream task defined in Section 3. We primarily focus on results from synthetic datasets where we have a unified question-generation procedure and access to underlying ground-truth factors. Our downstream models are transformer encoders with 2, 5, and 15 layers, which we refer to as T-$n$ with $n$ the number of layers. We train all combinations of upstream representation models and downstream classifiers, which amounts to 684 downstream models, with the cross-entropy loss.[1] We provide all implementation details in Appendix A and additional experimental results in Appendix D.

---

[1]Reproducing our experimental study requires approximately 13 GPU years on Nvidia A100 GPUs.

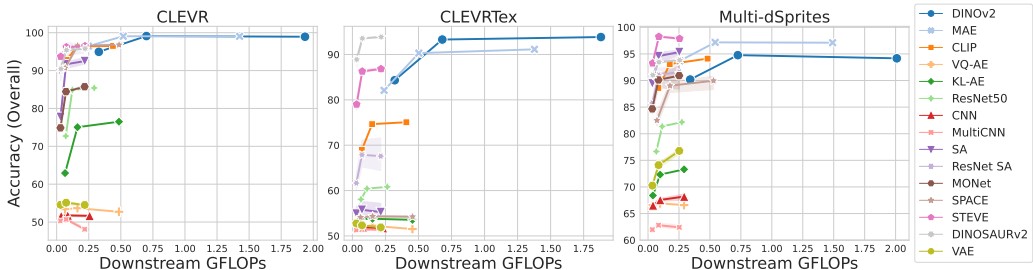

Figure 3: Average accuracies of different models vs. downstream GFLOPs across different datasets. Points along the x-axis represent T-2, T-5, and T-15, respectively. For pre-trained models, only 1 seed is available. For other models, the results are averaged over 3 random seeds and the shaded areas indicate 95% confidence intervals.

We carried out an extensive set of experiments with numerous baselines. To improve clarity, we have organized the results into key points and summarized the main takeaways. In the following, we report average results and confidence intervals over 3 random seeds, except for foundation models, where only 1 seed is available. We omit MONet's results on CLEVRTex due to its suboptimal performance, consistent with similar experimental results by Papa et al. (2022). When extending to real-world datasets, we keep only the pre-trained foundation models and top-performing OC models, excluding other upstream representation models due to their poor performance. Additionally, we report the results on VQA-v2 only with T-2 as the downstream model due to a degradation in performance observed when increasing the number of transformer layers (see Appendix A.3 for more details). Finally, unless explicitly mentioned, the Multi-dSprites version featured in the plots is the one comprising 320k unique images.

## 4.1    MAIN FINDINGS

**Performance of Large Foundation Models.**    Fig. 2 shows the overall accuracy for different upstream models across different synthetic datasets with T-15 as the downstream model, which generally achieves the best performance across synthetic datasets and upstream models. We observe that large foundation models, i.e., DINOv2, CLIP, and MAE, without any fine-tuning perform comparably well or the best on all datasets, although not by a large margin. However, when considering compute requirements, the picture appears more nuanced. In Fig. 3, which shows overall accuracy against the GFLOPs used for downstream training, we observe that some OC models achieve comparable performance to large foundation models with significantly less compute, making them more appealing under a limited compute budget.

It's important to emphasize the differences in model sizes and training data between foundation models and OC models. As shown in Table 2 in Appendix A.1, best-performing OC models like STEVE and ResNet SA are much smaller than their counterparts in the foundation model group and are specifically trained on the datasets studied in this work, which are significantly smaller than those used for training foundation models. Additionally, foundation models require substantial computational resources and significant engineering for pre-training, which are beyond our control. Therefore, carefully analyzing the effects of these factors in studies like ours is challenging.

**Effect of Object-Centric Bias.**    DINOSAURv2 (Seitzer et al., 2023), which consists of a pre-trained DINOv2 with Slot Attention applied downstream, allows us to explore the effect of applying the OC bias on a foundation model. Comparing the results of DINOv2 and DINOSAURv2 in Figs. 2 and 3 on CLEVRTex and Multi-dSprites, we observe that DINOSAURv2 outperforms DINOv2 while requiring significantly less downstream compute. However, on CLEVR, we do not observe the same patterns, as DINOSAURv2 performs slightly worse than DINOv2. After experimenting with different hyperparameter choices on CLEVR, we found that DINOSAURv2 is highly sensitive to these choices, which likely explains this suboptimal performance.

Additionally, by looking at Fig. 14 (Appendix D.3) which shows the overall accuracies on different downstream model sizes, we observe that on T-2, DINOv2 exhibits inferior performance compared to DINOSAURv2 on CLEVRTex, Multi-dSprites, and GQA. However, as we scale up the downstream model, starting from T-5, DINOv2 almost matches DINOSAURv2. This indicates that DINOv2 representations do contain the relevant information for the downstream task, but they seem to be less explicit and less readily usable, necessitating a larger downstream model compared to DINOSAURv2 to extract the required information effectively (Eastwood et al., 2023).

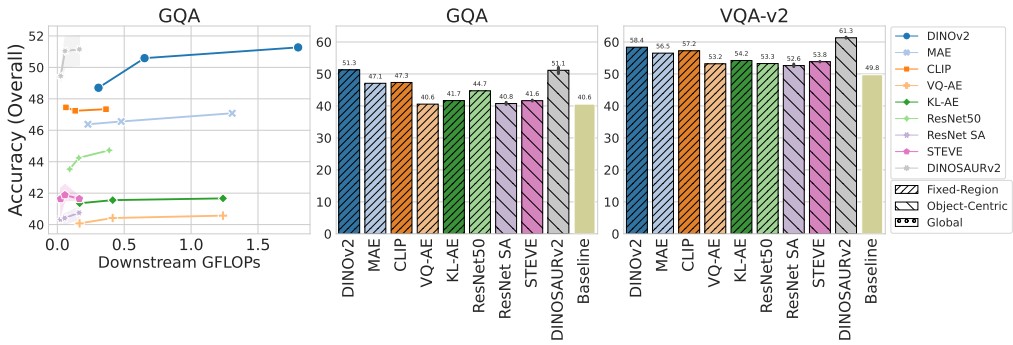

Figure 4: Left: Average accuracies of different models w.r.t. downstream GFLOPs on GQA. Points along the x-axis represent T-2, T-5, and T-15, respectively. For pre-trained models, only one seed is available. For other models, the results are averaged over 3 random seeds and the shaded areas indicate 95% confidence intervals. Middle & Right: Average accuracies on GQA and VQA-v2 for different upstream models, with T-15 and T-2 as the downstream models, respectively. The bars indicate means and 95% confidence intervals with 3 random seeds, when available.

**Performance of Other Upstream Models.** In Fig. 2, a discernible pattern emerges among upstream models. Generally, OC models consistently outperform other models except large foundation models. Smaller pre-trained models (VQ-AE, KL-AE, and ResNet50) tend to perform worse. Notably, on CLEVRTex, this trend is less pronounced, as most OC and pre-trained models struggle due to the dataset's complexity. Consistent with prior studies (Yoon et al., 2023), End-to-end CNN and MultiCNN models consistently score the lowest, and are followed by the global representation of VAEs. Additionally, on CLEVR and CLEVRTex, several models show only a slight improvement over the baseline, which relies solely on the question without any image-related information.

Within foundation models, DINOv2 and MAE consistently outperform others, with CLIP ranking as the third-best model probably due to its relatively smaller size. Looking at Table 2, we observe that while the good performance of DINOv2, MAE, and CLIP can likely be attributed to the size of their backbone, there appears to be no clear trend explaining the performance gap among smaller models.

**Real-world Data.** To investigate whether our findings hold in real-world scenarios, we conduct the same experiments on the GQA (Hudson & Manning, 2019) and VQA-v2 datasets (Goyal et al., 2017; Antol et al., 2015), two well-established benchmarks for the Visual Question Answering task. Fig. 4 left shows the overall accuracy of different upstream models on GQA vs. downstream GFLOPs, and the overall accuracy on GQA and VQA-v2 with T-15 and T-2 as the downstream models, respectively. Comparing DINOSAURv2 with DINOv2, we observe the same pattern as in synthetic datasets. Additionally, the performance trends across different models are consistent with those observed in synthetic datasets, further validating our primary conclusions and suggesting that the findings are robust across both real-world and synthetic datasets.

> **Takeaway.** While large foundation models can perform comparably to the best-performing OC models without any fine-tuning or hyperparameter adjustments, they generally require larger downstream models and more compute, presumably because their representations are less explicit than OC representations. On the other hand, the performance of many OC models drops on more complex datasets. Learning OC representations on top of a foundation model (see, e.g., DINOSAURv2) can be a viable solution to get the best of both worlds.

## 4.2 ADDITIONAL INSIGHTS

**Property Prediction vs VQA.** We additionally evaluate the representations on *property prediction*, a much simpler downstream task wherein the objective is to predict object properties from the representations. We adopt the same setup as Dittadi et al. (2022) (see Appendix A.4 for further details). In Fig. 5, we observe a strong correlation between accuracy of this simple task on most properties, and downstream VQA performance. This demonstrates that models capable of accurately predicting object properties excel on more challenging tasks like VQA. Therefore, performance on simple tasks like property prediction can be a useful evaluation metric for model selection. For the complete correlation results, see Appendix D.1.

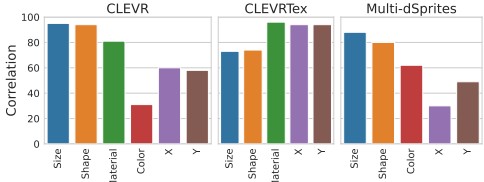 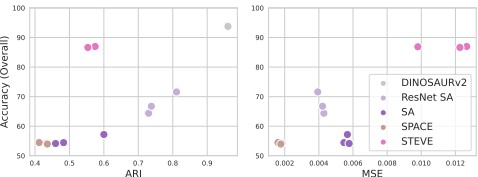

Figure 5: Correlation between property prediction accuracy (reported separately by object property) and overall VQA accuracy.

Figure 6: Overall VQA accuracy with T-15 as downstream model, plotted against ARI and MSE for different OC models on CLEVRTex.

> **Takeaway.** Overall, performance on object property prediction strongly correlates with VQA performance, indicating that much simpler tasks can be used to guide model selection.

**Upstream vs. Downstream Performance.** Fig. 6 depicts the relationship between upstream performance metrics and downstream VQA accuracy of OC models when using T-15 as the downstream model on CLEVRTex. Notably, STEVE exhibits the worst reconstruction MSE among OC models but achieves the second-best accuracy on VQA. This is not necessarily surprising: while Dittadi et al. (2022) observed a negative correlation between MSE and downstream performance, Papa et al. (2022) later showed this to no longer hold in the presence of textured objects. However, a higher ARI was shown to be predictive of better downstream performance. This appears not to hold in our case, as ResNet Slot Attention attains the second-best ARI but does not perform well in the VQA downstream task, while STEVE has a poor segmentation performance while achieving high accuracy. Further investigations are needed to shed more light on these trends, allowing for more robust upstream model selection strategies. For results on more upstream metrics and other datasets, see Appendix D.2.

> **Takeaway.** Upstream metrics such as ARI (segmentation) and MSE (reconstruction) are ***not*** good predictors of downstream performance on our VQA task.

**Effect of Training Size.** Fig. 7 shows the percentage decrease in the overall error rate of different models on Multi-dSprites when the training size increases from 40k to 320k unique images. Notably, with approximately 8x more data, most upstream models exhibit similar improvements, typically around 20–40%, regardless of their initial performance. CLIP is a notable exception, showing an increase in overall error rate of up to 50%. Additionally, end-to-end models (CNN and MultiCNN) and VQ-AE show only minimal improvement compared to the other models. See Appendix D.4 for further results, including raw accuracies and additional dataset sizes between 40k and 320k.

> **Takeaway.** Except for end-to-end models which show minimal improvements, all other models generally exhibit a similar performance gain with larger downstream training sizes.

**Consistency of the Results Across Question Types.** The average Spearman rank correlation between VQA accuracy on different question categories is 0.96 for CLEVR, 0.98 for CLEVRTex, 0.88 for Multi-dSprites, and 0.92 for VQA-v2. This suggests that the average VQA accuracy results shown in Figs. 2 and 4 are consistent across question categories. In Appendix D.5, Fig. 17 shows that for these datasets, the rank correlations are consistently high for all pairs of question categories, and Figs. 18 to 28 illustrate the complete results separately by question category. In contrast, on GQA, the correlation between accuracies of different question types is notably weaker. In particular, *Compare* and *Logical* questions exhibit low correlation with other categories. This is likely due to their reduced dependency on visual representations and heavier reliance on linguistic cues (Liu et al., 2022).

Delving deeper into the results for each category, it becomes apparent that on VQA-v2, *Number* questions, which require recognizing quantities in the image, are harder for all the models compared to *Yes/No* questions. On GQA, *Query* questions which demand complex, open-ended reasoning, consistently prove to be the most difficult, whereas *Verify* questions are the easiest. Moreover, models perform similarly on *Logical* and *Compare* questions, indicating that these question types depend much less on the visual information provided by the models. On synthetic datasets, we observe that *Count* questions, which necessitate an understanding of the existence of multiple objects with specific properties, are generally the most challenging for almost all models. In contrast, *Exist* questions are the easiest, which is expected because they check for the existence of a single object with specific properties. Among *Compare Integer* questions, *Equal* questions appear to be the most challenging,

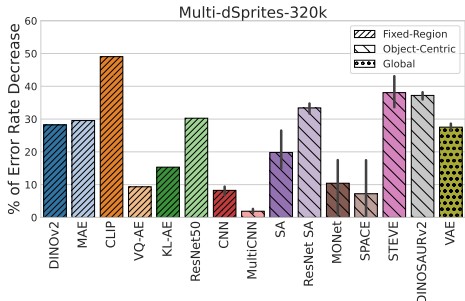
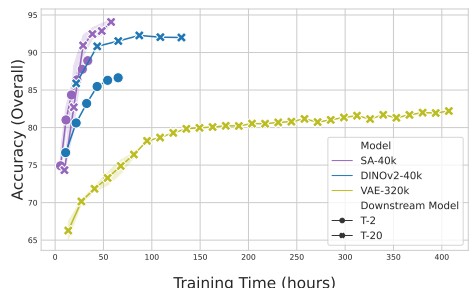

Figure 7: Average % decrease in VQA error rate (means and 95% CIs with 3 random seeds) for different upstream models on Multi-dSprites, when increasing the training set size from 40k to 320k, using T-15 as the downstream model.

Figure 8: VQA accuracy vs. downstream training time on Multi-dSprites. Slot Attention and DINOv2 are trained on 40k images, while the VAE is trained on 320k images for 3 million steps (5x more than the other models).

requiring an exact count of two sets of objects. Finally, in *Attribute* questions, *Size* questions emerge as the easiest, while there is no specific discernible pattern among other object attributes.

> **Takeaway.** While some question categories are on average more difficult than others, we observe a strong correlation between accuracies across different question categories.

**Evaluation of Global Representations.** Here we further investigate whether the global representations of a VAE can match the performance of OC representations when given a significant advantage in terms of data and training budget. For this purpose, we train a downstream transformer encoder model with 20 layers (T-20) on top of the VAE on Multi-dSprites with 320k unique images for 3 million steps and compare the result with other models trained on Multi-dSprites with the smallest training size (40k), with the smallest downstream model (T-2) trained for the default number of training steps (600k). From Fig. 8, it is evident that the performance of T-20 trained on top of the VAE cannot match the performance of T-2 trained on top of Slot Attention and DINOv2. In conclusion, even with a larger downstream model, more training steps, and a larger training dataset size, global representations of VAEs cannot match the performance of OC models and therefore do not seem ideal for downstream tasks related to objects.

> **Takeaway.** Even with significantly more training data and compute, global representations such as those of standard VAEs are far from competitive. This corroborates the common assumption that such representations are not suitable for object-related downstream tasks.

## 5  CONCLUSION AND DISCUSSION

In this study, we systematically assess OC representations on downstream reasoning tasks by comparing them with foundation models and various benchmarks across three synthetic and two real-world multi-object datasets. Our primary focus is the VQA task, which requires a precise compositional understanding of images, objects, and their relationships. Our findings indicate that foundation models perform comparably to OC models without requiring fine-tuning or hyperparameter adjustments. However, they are significantly larger and demand greater computational resources. Overall, this points to a complex trade-off between model classes. Still, one can benefit from both worlds by applying the OC bias to foundation models. Beyond performance comparisons, we emphasize the importance of downstream evaluations, as they provide a more pragmatic assessment of representation quality and align well with the goals of OC learning—particularly in capturing the compositional properties of scenes. We encourage future research to adopt such evaluations to better understand the strengths and limitations of different approaches.

While our study covers commonly used OC datasets, they are limited to static images. Future work could extend these analyses to video data, where dynamic scene understanding presents additional challenges. Additionally, further research could explore the effects of fine-tuning foundation models, both with and without OC inductive biases, as well as systematically study the generalization capabilities of OC models in other tasks, such as causal inference.

ACKNOWLEDGMENTS

We would like to thank Thomas Kipf, Max Horn, and Sindy Löwe for the helpful discussions and comments. This work was partially supported by the Wallenberg AI, Autonomous Systems and Software Program (WASP), funded by the Knut and Alice Wallenberg Foundation, and by the Helmholtz Foundation Model Initiative, supported by the Helmholtz Association. A.D. acknowledges support from G-Research. The computations were enabled by the Berzelius resource, provided by the Knut and Alice Wallenberg Foundation at the National Supercomputer Centre, and by the Gauss Centre for Supercomputing e.V. (www.gauss-centre.eu), which provided the required computing time through the John von Neumann Institute for Computing (NIC) on the GCS Supercomputer JUPITER | JUWELS (Jülich Supercomputing Centre, 2021) at Jülich Supercomputing Centre (JSC).

ETHICS STATEMENT

In this work, we are taking a step in the direction of systematically analyzing and understanding the reasoning capabilities of deep learning systems with a particular focus on object-centric models, which benefits both the research community and society. We do not see a negative societal impact of this work beyond what is brought about by general advances in machine learning.

REPRODUCIBILITY STATEMENT

All reproduction details regarding the models, hyperparameters, implementation, and training and evaluation procedures are provided in the main text and appendix. Specifically, information about the upstream models, downstream VQA pipeline, and downstream property prediction pipeline is available in Appendix A. We use publicly available implementations of the models, which are properly referenced in the paper. All datasets are publicly available, and details about each dataset, as well as the question generation procedure, can be found in Appendix B. Finally, the metrics are described and properly referenced in Appendix C.

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

## A MODELS AND IMPLEMENTATION DETAILS

Here we elaborate on the upstream and downstream models included in this study along with details on the training, the implementation, and hyperparameter choices.

### A.1 UPSTREAM MODELS

Here we elaborate on all the upstream models we use in our experiments and provide details on the implementation, training, and hyperparameter choices.

**Implementation & Training Details.** Our code is based on the implementations of object-centric models of Dittadi et al. (2022) and we use their implementation of Slot Attention, MONet, SPACE, and VAE. For these models, we use the same set of recommended hyperparameters on CLEVR and Multi-dSprites, and apply the same hyperparameters for CLEVRTex as used in CLEVR. All other models are either re-implemented or adapted from available code. Unless explicitly stated otherwise, except for pre-trained foundation models, we train all the other models with a default batch size of 32 for 3 random seeds with Adam optimizer. The training finishes after 500k steps on synthetic datasets and 250k steps on real-world datasets. The reported metrics are then averaged over the seeds to provide a comprehensive assessment. Additionally, more information about pre-trained models and the size of the models are shown in Table 2. The details of each model are explained below.

**DINOv2.** DINOv2 (Oquab et al., 2023), an enhanced version of the DINO (Caron et al., 2021), stands out as a self-supervised ViT-based model designed for training high-performance computer vision models without the need for extensive labeled data. It serves as a versatile backbone for diverse tasks, including image classification, video action recognition, semantic segmentation, and depth estimation. Trained on a carefully curated dataset comprising 142 million images with a discriminative self-supervised method, DINOv2 excels in producing versatile visual features that transcend specific image distributions and tasks without the necessity for fine-tuning.

Similar to DINO, DINOv2 follows a transformer architecture, with a patch size of 14, and is trained with 1B parameters with a self-supervised learning objective, and distilled into a series of smaller models that generally surpass the other best available all-purpose features. There are 4 distinct backbone versions, each varying in the number of transformer layers. After experimenting with all 4, considering a balance between the performance and downstream training time, we selected the second-largest variant, denoted as *ViT-L/14*. We employ this backbone without fine-tuning, and uniformly resize images from all datasets to dimensions of $224 \times 224$ and pass them to the model, generating fixed-region $16 \times 16$ representations with a channel size of 1024 for each patch. We flatten the spatial dimensions and pass a matrix of size $256 \times 1024$ to the downstream model.

**MAE.** The Masked Autoencoder (MAE) (He et al., 2022) is a simple approach for reconstructing an original signal from a partial observation. In this approach, the image is divided into non-overlapping patches, with 75% of them randomly masked. These patches are then fed into a ViT-based encoder, which converts the partially observed input into a latent representation. Next, a lightweight decoder, which uses the representation and mask tokens, reconstructs the original image. The model is trained on ImageNet-1K (Deng et al., 2009) by minimizing the Mean Squared Error (MSE) between the reconstructed and original input in the pixel space.

There are various pre-trained MAEs available, differing in model sizes. In our framework, we utilize the pre-trained *ViT-L/16*. We resize the input images to $224 \times 224$ and pass them to the encoder without masking. This generates a fixed-region representation sequence of length 197 (196 corresponding to different regions in the image and one corresponding to the output of the *CLS* token) with a latent size of 1024 for each sequence. We pass the obtained representation matrix without any modifications to the downstream model.

**CLIP.** The Contrastive Language-Image Pretraining (CLIP) (Radford et al., 2021) is a multimodal model that learns to associate images and corresponding text descriptions. CLIP uses a ViT to extract a feature vector that represents the visual content of the image. Similarly, for the text, CLIP uses a transformer-based model to generate a feature vector that represents the semantic content of the text. These two feature vectors are then projected into a shared embedding space and the cosine similarity between the two vectors is calculated. The model is trained on a dataset of 400 million (image, text) pairs collected from a variety of publicly available sources on the Internet, using a contrastive loss function which is a symmetric cross-entropy loss over the similarity scores, that encourages the

Table 2: Model sizes and training information of the models.

| Model | Model Architecture | # of Params[†] | Pre-training Dataset | Dataset Size |
|---|---|---|---|---|
| DINOv2 | ViT-L/14 | 304M | LVD-142M | 142M |
| MAE | ViT-L/16 | 303M | INet-1k | 1.2M |
| CLIP | ViT-B/32 | 87M | WIT-400M | 400M |
| VQ-AE | - | 34.5M | OpenImages v4 | 9M |
| KL-AE | - | 34.5M | OpenImages v4 | 9M |
| ResNet50 | - | 23M | INet-21k | 14M |
| CNN | - | 0.1M | - | - |
| MultiCNN | - | 1.4M | - | - |
| SA | - | 0.9M | - | - |
| ResNet SA | - | 22.4M | - | - |
| MONet | - | 1.6M | - | - |
| SPACE | - | 5.3M | - | - |
| STEVE | - | 17.1M | - | - |
| DINOSAURv2[*] | - | 4M + 304M | - | - |
| VAE | - | 19.2M | - | - |

[*]# of parameters of pre-trained DINOv2 is shown separately.
[†] Except for foundation models, # of parameters are reported on CLEVR.

similarity between the image and text feature vectors to be high when they are a matching pair and low when they are not.

In our experiments, we utilize a pre-trained CLIP image encoder with a ViT architecture denoted as *ViT-B/32*. Similar to DINOv2 and MAE, we resize the input images for all datasets to $224 \times 224$ and pass them to the encoder. The encoder produces a representation of size $50 \times 768$ (one corresponding to the output of the *CLS* token and the rest corresponding to regions in the image) which we utilize directly in the downstream model.

**VQ-AE & KL-AE.** VQ-AE and KL-AE are two pre-trained autoencoders of the latent diffusion model (LDM) (Rombach et al., 2022). In LDM, they don't directly use a diffusion model in pixel space. Instead, to facilitate training on constrained computational resources without compromising quality and flexibility, they apply the models in the latent space of a powerful autoencoder pre-trained on OpenImages (Kuznetsova et al., 2018) with 9M images in an adversarial manner. The autoencoder consists of an encoder that downsamples the images by a factor $f$, and a decoder that reconstructs the original image. In order to avoid arbitrarily high-variance latent spaces, they apply two different regularizations: one imposes a KL penalty towards a standard normal on the learned latent (KL-AE), similar to a VAE, and the other one uses a vector quantization layer (Van Den Oord et al., 2017) within the decoder (VQ-AE).

Various pre-trained autoencoders with different downsampling factors are available. In our experiments, we explore multiple models with varying downsampling factors and find that a factor of 16 is the balancing point between performance and training speed. Therefore, we adopt models with this factor for further analysis. In our experiments, we utilize the encoder of these two autoencoders off-the-shelf. By applying the encoders, we get a vector of size $W/16 \times H/16 \times D_{fr}$ where $D_{fr}$ is 16 and 8 for KL-AE and VQ-AE, respectively. Similar to DINOv2, we flatten 2d feature maps and feed them into the downstream model.

**ResNet50.** ResNet50 He et al. (2016) is a deep neural network with 50 layers that has been pre-trained on ImageNet-21k (Deng et al., 2009). It is well-known for its residual learning blocks and serves as a baseline in our study. In our experiments, we employ the off-the-shelf ResNet50 model and remove the pooling and fully connected layers at the end. We apply the default ResNet50 transformations on the input image, pass it to the model, and obtain a vector of size $W/32 \times H/32 \times 2048$ which we flatten the spatial dimensions and pass to the downstream model.

**CNN.** CNN (Zambaldi et al., 2018) is a small convolutional neural network that is commonly used in the literature. It consists of a few convolutional layers with ReLU activation functions in between, and is trained end-to-end with the downstream model on the downstream task. It produces a fixed-size

Table 3: Hyperparameters of CNN.

| CNN | | | | |
|---|---|---|---|---|
| **Dataset** | **Kernel Size** | **Stride** | **Output Channels** | **Activation Function** |
| CLEVR & CLEVRTex | 8 | 4 | 32 | ReLU |
| | 4 | 2 | 64 | ReLU |
| | 4 | 2 | 64 | ReLU |
| | 3 | 1 | 64 | ReLU |
| Multi-dSprites | 8 | 4 | 32 | ReLU |
| | 4 | 2 | 64 | ReLU |
| | 3 | 1 | 64 | ReLU |

fixed-region representation of size $4 \times 4 \times 64$ which we flatten the spatial dimensions and produce a vector of size $16 \times 64$ and pass it to the downstream model. The hyperparameters of CNN are shown in Table 3.

**MultiCNN**   MultiCNN (Kipf et al., 2019) is an object-centric model consisting of $N_{slots}$ CNNs, each dedicated to detecting a single object within an image. These CNNs operate with non-shared parameters, processing each input image independently. MultiCNN shares the same dataset-specific hyperparameters as the CNN baseline, and similar to CNN, it is trained end-to-end using the same training hyperparameters and loss function as the downstream model. The output of each CNN undergoes complete flattening, followed by a shared linear layer of size 64, resulting in a representation of size $N_{slots} \times 64$, which is then passed to the downstream model.

**Slot Attention.**   Slot Attention (Locatello et al., 2020) has become the primary representative for object-centric (OC) learning in recent years. It follows an autoencoder setup and begins with a Convolutional Neural Network (CNN) and is followed by the Slot Attention module. This module refines the initial image features through multiple iterations, turning them into distinct slots representing objects. Each slot is updated using a Gated Recurrent Unit (GRU) that takes the current slot and attention information as inputs. After refining, these slots are used to reconstruct the appearance and mask of each object, which are then combined to reconstruct the original image. The model is trained by minimizing the MSE reconstruction loss.

We also employ an improved version of Slot Attention introduced in Biza et al. (2023) which we refer to as *ResNet SA*. In this improved version, the CNN backbone is replaced by a ResNet34 (He et al., 2016) without pre-training, and a larger feature map resolution of 16 on synthetic datasets and 7 on VQA-v2 and GQA is used in both the encoder and the decoder of the model. Furthermore, the initial slots are changed into learnable slots. Both the original and improved models are trained on each dataset with a batch size of 64, a learning rate of 0.0004, a learning rate warmup of 10k steps, and an exponential learning rate decay with a half-life of 100k steps. For ResNet SA, we additionally clip the gradient norm at 0.05 to stabilize training. We follow the same architecture for ResNet SA as in Biza et al. (2023) on synthetic datasets. On VQA-v2 and GQA, we modify the architecture by replacing the initial convolutional layer of ResNet34 with a convolutional layer featuring a kernel size of $7 \times 7$ and a stride of 4. Additionally, in the decoder, we incorporate two additional transpose convolutional layers at the beginning, each configured with the same hyperparameters as the existing transpose convolutional layers. After training, the learned slot vectors of size $N_{slots} \times 64$ are used as representations in the downstream model for both versions.

**MONet.**   The Multi-Object Network (MONet) (Burgess et al., 2019) consists of a recurrent segmentation network that generates attention masks that represent the probability of each pixel belonging to each object. For each slot, a VAE (the component VAE) encodes the image and the current attention mask, and decodes the latent representation to an image reconstruction of the slot and the slot mask. To create the final reconstructed image, the reconstructed images are combined using the attention masks obtained from the segmentation network. The model is trained by an objective function comprising a reconstruction loss defined as the negative log-likelihood of a spatial Gaussian mixture model (GMM) with one component per slot, where each pixel is modeled independently, and a KL divergence of the component VAE, and an additional mask reconstruction loss for the component VAE. The mean of the GMM for each component is used as the representation of each object. The learned representation is a vector of size $N_{slots} \times 16$ and is directly used for the downstream task.

Table 4: Hyperparameters of STEVE.

| STEVE | | |
|---|---|---|
| **Module** | **Hyperparameter** | **Hyperparameter Value** |
| Encoder | Corrector Iterations | 2 |
| | Slot Size | 192 |
| | MLP Hidden Size | 192 |
| | # Predictor Blocks | 1 |
| | # Predictor Heads | 4 |
| | Learning Rate | 0.0001 |
| Transformer Decoder | # Decoder Blocks | 8 |
| | # Decoder Heads | 4 |
| | Hidden Size | 192 |
| | Dropout | 0.1 |
| | Learning Rate | 0.0003 |
| DVAE | Learning Rate | 0.0003 |
| | Patch Size | $4 \times 4$ pixels |
| | Vocabulary Size | 4096 |
| | Temperature Start | 1.0 |
| | Temperature End | 0.1 |
| | Temperature Decay Steps | 30k |

**SPACE.** Spatially Parallel Attention and Component Extraction (SPACE) (Lin et al., 2020b) provides a unified probabilistic modeling framework that combines the best of spatial attention and scene-mixture approaches. Foreground objects are identified using bounding boxes computed in a parallel spatial attention process, and background elements are modeled using a mixture of components. The model is trained by optimizing the Evidence Lower Bound (ELBO) of the probabilistic model. An additional boundary loss is introduced to penalize the splitting of objects across bounding boxes, addressing potential under- or over-segmentation issues. SPACE representations are vectors of size $69 \times 38$ where 69 is the number of slots that is determined by the grid size, and the latent representation of each slot is obtained by concatenating all the latent variables which will have a dimension of 38. We utilize this representation directly in the downstream model.

**STEVE.** STEVE (Singh et al., 2022b) is a simple object-centric video model achieving remarkable performance over various complex and naturalistic videos. It is a more robust version of SLATE (Singh et al., 2022a), a state-of-the-art object-centric model. The model contains two reconstruction paths. The first path uses a discrete VAE encoder to convert the input image into discrete tokens, and then a discrete VAE decoder to reconstruct the original image. This path is trained using MSE reconstruction loss. The second path uses a CNN-based image encoder on the input, and the output is fed into a recurrent slot encoder that updates slots over time using recurrent neural networks. Finally, a slot-transformer decoder, similar to SLATE, is applied to the produced slots to predict and reconstruct the discrete tokens of the input image. This path is trained by minimizing the cross-entropy loss between the original tokens produced by the discrete VAE encoder, and the predicted tokens of the slot-transformer decoder.

We incorporate the original implementation of STEVE into our framework. STEVE works on videos, considering the input to be a sequence of $t$ image frames. Following the authors' recommendation, to utilize it on images, we treat images as 1-frame videos and pass them to the model. On synthetic datasets, we train STEVE for 500k steps with an exponential learning rate decay with a half-life of 250k steps and with 30k warm-up steps. For , we employ the same training hyperparameters but reduce the number of steps to 250k. After the training, we use slots of size $N_{slots} \times 192$ obtained from the recurrent slot encoder in the downstream model.

A summary of the model's hyperparameters on all datasets is shown in Table 4. We maintain the original architecture for the CNN backbone and discrete VAE encoder/decoder for synthetic datasets. However, for VQA-v2 and GQA with image sizes of $224 \times 224$, we reduce the feature map dimensions of both the discrete VAE encoder and CNN backbone by half. This adjustment includes changing the stride of the first convolutional layer of the CNN backbone to 4. Additionally, in the discrete

Table 5: Hyperparameters of DINOSAURv2.

| **DINOSAURv2** | | | | |
| --- | --- | --- | --- | --- |
| **Hyperparameter** | | **CLEVR & Multi-dSprites** | **CLEVRTex** | **VQA-v2 (COCO) & GQA** |
| Training Steps | | 250k | 500k | 250k |
| Batch Size | | 64 | 64 | 64 |
| LR Warmup Steps | | 10k | 10k | 10k |
| Peak LR | | 0.0004 | 0.0004 | 0.0004 |
| Exp. Decay Half-Life | | 100k | 100k | 100k |
| ViT Architecture | | ViT-L | ViT-L | ViT-L |
| Patch Size | | 14 | 14 | 14 |
| Feature Dim. | | 1024 | 1024 | 1024 |
| Gradient Norm Clipping | | 1.0 | 1.0 | 1.0 |
| Image Size | | 224 | 224 | 224 |
| Cropping Strategy | | Full | Full | Full |
| Image Tokens | | 256 | 256 | 256 |
| Decoder | Type | MLP | MLP | MLP |
| | Layers | 4 | 4 | 4 |
| | MLP Hidden Dim. | 1024 | 512 | 2048 |
| Slot Attention | Iterations | 3 | 3 | 3 |
| | Slot Dim. | 256 | 256 | 256 |
| | MLP Hidden Dim. | 1024 | 1024 | 1024 |

VAE encoder, we insert a convolutional layer with a $2 \times 2$ kernel and stride 2, followed by ReLU activation, after the initial convolutional layer. To accommodate these feature map changes in the discrete VAE decoder, we duplicate the four convolutional blocks and the pixel shuffling layer before the final convolutional block.

It is noteworthy that the training of STEVE is not entirely stable, as we observed that upstream performance metrics begin to degrade after some training time (after roughly 20-40k steps). However, when training the model for longer, we observe that it performs well on the downstream task. Nevertheless, in some seeds, the training fails as the upstream metrics are significantly worse than in other seeds. Therefore, we only consider the seeds that are stable and perform the best in terms of reconstruction and segmentation quality.

Additionally, we experimented with a modified version of STEVE using a pre-trained DINOv2 as the CNN encoder backbone. However, it performed similarly or worse than the original STEVE on the downstream VQA. We suspect this may be due to the chosen hyperparameters or an architectural bottleneck in the discrete VAE. As a result, we decided not to include it in our reported results.

**DINOSAUR.** DINO and Slot Attention Using Real-world data (DINOSAUR) (Seitzer et al., 2023) is an object-centric model designed to bridge the gap between object-centric models and real-world data. It consists of an encoder that extracts features from the input data, a slot attention module that groups the extracted features into slots, and a decoder that reconstructs the extracted features. Their approach can be considered similar to SLATE (Singh et al., 2022a) and STEVE (Singh et al., 2022b), but with the difference of reconstructing global features from a pre-trained Vision Transformer (Dosovitskiy et al., 2020) instead of local features from a VQ-VAE (Van Den Oord et al., 2017).

We adapt the original implementation of DINOSAUR into our framework and replace the pre-trained DINO backbone with pre-trained DINOv2 (Oquab et al., 2023), which we refer to as DINOSAURv2. Similar to the original training procedure of DINOSAUR, the input images are resized to $224 \times 224$ and we train DINOSAURv2 for 500k steps on CLEVRTex, and 250k steps on CLEVR, Multi-dSprites, COCO images of VQA-v2, and GQA images. The training uses a learning rate of 0.0004, a learning rate warm-up of 10k optimization steps, and an exponentially decaying learning rate schedule. Furthermore, we clip the gradient norm at 1 to stabilize training. After the training, we use the slots of size $N_{slots} \times 256$ obtained from the Slot Attention module in the downstream model. The full hyperparameters of the model are provided in Table 5.

**VAE.** We train a variational autoencoder (VAEs) (Kingma et al., 2014) with a broadcast decoder (Watters et al., 2019) as a baseline that learns global representations. We use the broadcast VAE implementation of Dittadi et al. (2022) with the same architecture and hyperparameter choices. For CLEVRTex, we use the same hyperparameters as in CLEVR. The latent size is selected to be 64 times the number of slots used when training an object-centric model on the same dataset. As explained in Section 3.5, we divide the flat representation vector into $N_{slots}$ vectors of size 64, and pass them to the downstream model.

## A.2 TEXT EMBEDDING MODULE

As our text embedding module, we use Text-to-Text Transfer Transformer (T5) (Raffel et al., 2020), a transformer-based language model developed by Google AI Language. It is capable of performing a wide range of natural language processing tasks such as text classification, question answering, summarization, and translation. The model is trained on the colossal, cleaned version of Common Crawl's web crawl corpus (C4), an 806-gigabyte corpus of text data using a pretext task called Text-to-Text Transfer Transformer (T5), which involves converting a given input text to a target output text. We utilize the available implementation in Hugging Face's Transformers library. T5 comes in different sizes and we use the T5-base tokenizer and encoder which produces the representations of size 768 for each token.

## A.3 DOWNSTREAM VQA SETUP

**Architecture and Hyperparameters.** We use a transformer-based architecture (Vaswani et al., 2017) which is the standard downstream architecture for the VQA task (Ding et al., 2021a; Devlin et al., 2018; Lu et al., 2019). We utilize the original PyTorch implementation of the transformer. We first project the image and the text representations with two separate linear layers with a 126 size and a dropout of 0.1. Then, we augment the image and text vectors with a 2-dimensional one-hot vector indicating whether the input is from the image representation or the text embeddings. A sinusoidal positional encoding is then added to the text embeddings. We introduce a trainable vector $CLS \in \mathbb{R}^{128}$, akin to the *CLS* token in *BERT* (Devlin et al., 2018), to generate classification results. The image and text representations, along with the *CLS* token, are concatenated and passed through a transformer encoder with a d_model of 128 and a hidden dimension of 128. The transformed value of the *CLS* token is passed through a classifier MLP that generates a probability vector over all possible answers in each dataset. The MLP consists of 2 linear layers of size 128. A normalization layer, a dropout of 0.1, and a ReLU activation function are applied in between the layers.

**Training.** On synthetic datasets, all downstream models are trained with a batch size of 128 and a learning rate of 0.0001 for 600k training steps, with the cross-entropy loss. However, when using DINOv2 and MAE as upstream models, it is infeasible to keep the current batch size with only one GPU due to the substantial sequence length of the representations. To ensure a fair comparison, gradients are accumulated, and the optimizer is applied every 4 training step with a reduced batch size of 32. Consequently, the downstream model is trained for 2.4 million steps, 4 times the default number of steps.

On GQA, the downstream models are trained for 900k steps with a batch size of 32 and a learning rate of 0.0001, using the same loss function as for synthetic datasets. On VQA-v2, we train all downstream models with the same batch size and loss function for 300k steps. Furthermore, on VQA-v2, we use a learning rate of 0.0001 for T-2 and T-5, and 0.00005 for T-15. However, we find that T-2 outperforms the other downstream models, with performance degrading significantly as the number of layers increases which is due to overfitting caused by the much smaller training size of VQA-v2 compared to other datasets in our study. As a result, we only report results for T-2 on VQA-v2.

## A.4 DOWNSTREAM PROPERTY PREDICTION SETUP

Here we assess representations by training downstream models to predict ground-truth object properties from these representations. Following the approach outlined by Dittadi et al. (2022) for object property prediction on synthetic datasets. in summary, we employ a single downstream model $f$ to predict the properties of each object independently. To be more specific, for OC models, we apply $f$ on each slot representation, i.e. $\hat{y}_k = f(z_k)$, where $z_k$ is the $k$th slot representation and $\hat{y}_k$ is the predicted properties of the $k$th slot. Similarly, for models with fixed-region representations, we treat each region as a slot and apply the same approach as with OC models. We find this approach to be effective for these models. Lastly, for models with global representations, since the representations of

individual objects are not readily available, we adopt the same strategy as demonstrated in Dittadi et al. (2022) which has proven to be working. We predict the properties of all objects using the downstream model $f$ and split it into $K$ vectors $\{\hat{y}_k\}_{k=1}^K$ where $K$ roughly corresponds to the number of slots in an OC model.

Since the predicted properties might not correspond to the objects in the same order as the ground-truth objects, and the number of slots can exceed the number of objects in the image, we follow Dittadi et al. (2022); Locatello et al. (2020) in using the same loss-matching algorithm to match $\hat{y}_k$ with its corresponding ground-truth vector. We define $f$ as an MLP with 1 hidden layer of size 256, and we utilize cross-entropy loss for categorical properties and MSE for numerical properties. We train $f$ for one seed on 10000 images of each dataset using Adam optimizer with a learning rate of 0.001 and a batch size of 64 for 6000 steps. For a randomly selected test set of 2000 images, we calculate the accuracy for categorical properties and the adjusted $R^2$ for numerical properties and report the results.

## B    DATASETS

We work with 3 existing multi-object datasets: *Multi-dSprites*, *CLEVR*, and *CLEVRTex*. We use the common format for these datasets as outlined in Dittadi et al. (2022). We primarily focus on these datasets due to their unified question-generation procedure and complete access to all underlying ground-truth object properties. Additionally, we utilize *VQA-v2* and *GQA* to extend our study to real-world settings. All datasets are summarized in Table 6. More details about each dataset are provided in the next subsections.

### B.1    OVERVIEW OF DATASETS

**CLEVR**    *CLEVR* (Johnson et al., 2017) comprises $128 \times 128$ images depicting 3D scenes with a plain gray background featuring up to 10 objects that may partially occlude one another. Objects vary in colors (8 options in total), materials (rubber or metal), shapes (sphere, cylinder, cube), sizes (small or large), and positions (x and y) as well as rotations. Following previous works (Locatello et al., 2020; Dittadi et al., 2022; Greff et al., 2019), we utilize the CLEVR6 variant to learn object-centric representations in which the number of objects is limited to 6. The dataset has been cropped and resized according to the procedure detailed originally by Burgess et al. (2019).

**CLEVRTex**    The *CLEVRTex* dataset extends the original *CLEVR* (Johnson et al., 2017) by incorporating textures on object surfaces, introducing a more visually complex environment. Each scene in *CLEVRTex* consists of 3-10 objects with distinct shapes (cube, cylinder, sphere, monkey head), sizes (small, medium, large), and textures (60 in total), contributing to the diversity of visual features. The backgrounds also present complex textures compared to the plain gray ones in CLEVR. The dataset is designed to facilitate the exploration of models' abilities in handling textured objects, providing a valuable resource for evaluating the performance of vision-related tasks in the context of rich visual scenes.

**Multi-dSprites**    This dataset is derived from the dSprites dataset (Matthey et al., 2017) of $64 \times 64$ synthetic images. Following prior research (Dittadi et al., 2022; Locatello et al., 2020; Greff et al., 2019), we utilize the Multi-dSprites variant, featuring colored sprites set against a grayscale background where the intensity of the uniform grayscale background is randomly determined for each image. Each scene consists of 2–5 objects with randomized attributes, including shapes (ellipse, square, heart), sizes (selected from 6 discrete values in $[0.5, 1]$, and converted to small and large with a threshold of 0.8), x and y positions, orientation, and color (randomly sampled in HSV space). Objects might occlude one another, with certain objects being nearly entirely concealed by others in specific images. Consequently, we eliminate images where an object is significantly obscured by another object, leaving only those with clearer visibility of individual objects. We use 4 different training sizes of this dataset which is demonstrated in Table 6.

**VQA-v2**    *VQA-v2* (Goyal et al., 2017; Antol et al., 2015) pairs open-ended questions with images from the MS COCO 2014 dataset (Lin et al., 2014). It serves as a benchmark for evaluating how well models can comprehend and reason about visual information in real-world scenarios. The questions cover a wide range of topics and require a detailed understanding of the image content, spanning from basic inquiries about object presence to more complex queries about relationships and attributes within the scene. Answers are categorized into *Yes/No*, *Number*, and *Other* types. Each question has 10 ground-truth answers, with the most frequent ground-truth answer considered correct in our

Table 6: Dataset splits for upstream and downstream training, number of slots used in training slot-based object-centric models, and count of unique answers to questions in each dataset.

| Dataset Name | Slots* | Answers | Dataset Splits† | | |
|---|---|---|---|---|---|
| | | | Train Size | Validation Size | Test Size |
| CLEVR6 | 7 | 24 | 2314980 (49000) | 70702 (1500) | 70997 (1500) |
| CLEVRTex | 11 | 20 | 1489005 (40000) | 55977 (1500) | 55646 (1500) |
| Multi-dSprites‡ | 6 | 22 | 1545444 (320000) | 57816 (1500) | 57557 (1500) |
| | | | 3089425 (320000) | 57540 (1500) | 57557 (1500) |
| | | | 6181799 (320000) | 58289 (1500) | 57557 (1500) |
| | | | 12365042 (320000) | 57792 (1500) | 57557 (1500) |
| VQA-v2 (COCO) | 7 | 17 | 215553 (82753) | 10000 (10000) | 103717 (40504) |
| GQA | 7 | 1834 | 943000 (72140) | 132044 (10234) | 12576 (398) |

* We define it as the maximum number of objects plus one additional slot for the background.
† The values in parentheses denote the size of the corresponding image split used for the upstream model.
‡ For Multi-dSprites, We use 4 different training sizes.

evaluation framework. We filter out specific questions in the dataset based on their answer type, as explained in detail in Appendix B.2. We train downstream models using the filtered *train* split of the dataset. However, since answer types for the *test* split are not available, we cannot apply the same filtering process to evaluate on this split properly. Therefore, we utilize the *val* split as our test set for evaluation, and additionally select a subset of 10k questions from it as our validation set during training.

The *MS COCO* (Microsoft Common Objects in Context) dataset (Lin et al., 2014) is a widely used collection of images designed for object detection, segmentation, and captioning tasks. It contains images sourced from everyday scenes that are diverse in content, encompassing various objects, activities, and environments. The dataset includes over 200k labeled images across 80 object categories, such as people, animals, vehicles, and indoor objects, with a significant emphasis on diversity in scenes and object appearances. Following Seitzer et al. (2023), we resize all the images to $224 \times 224$ without any cropping, while ignoring the aspect ratio. We use the original split of the dataset when training and evaluating the upstream models. Furthermore, during upstream training, we augment the dataset by horizontally flipping images with a probability of $0.5$.

**GQA** *GQA* (Hudson & Manning, 2019) is a benchmark for visual reasoning, featuring open-ended questions about images that require a detailed understanding of object relationships and spatial reasoning. it provides both questions and corresponding scene graphs, making it particularly suited for evaluating relational and compositional reasoning in multi-object environments. This makes GQA a challenging and insightful benchmark for assessing object-centric and foundation models alike. The dataset is available in both balanced and unbalanced versions. The balanced version ensures a uniform distribution of answers across question categories, focusing on reasoning over memorization, while the unbalanced version reflects more natural distributions of answers. In our work, we utilize the balanced version, with the *train* split used for training and the *testdev* split used for evaluation. Furthermore, we use the same image preprocessing and augmentations as for VQA-v2.

### B.2 QUESTION GENERATION & PREPROCESSING

**Synthetic.** Originally, excluding CLEVR, the synthetic datasets exclusively comprise images without associated questions. Consequently, for their transformation into Visual Question Answering (VQA) datasets, we employ a question generation mechanism based on Johnson et al. (2017), adapted for all datasets. The process involves utilizing a script that takes as input a JSON file that contains scene information and object features in the dataset. The script outputs a JSON file containing questions for each image. To generate questions for each image, there are 9 different question templates that take the features of the objects in each image and generate a question based on the template. These templates are adjusted for each dataset, producing a maximum of 50 questions for each image in each dataset. However, due to some images featuring only a few objects, the average number of questions per image tends to be less than 50 in each dataset. Additionally, we refrain

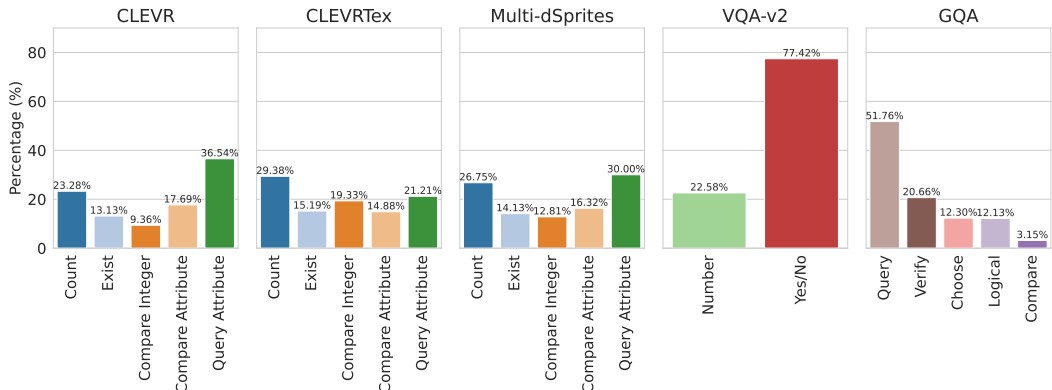

Figure 9: Distribution of question categories per dataset.

from utilizing pre-existing questions for CLEVR. Instead, we generate questions for it to establish a standardized question-generation process for all datasets.

Question templates can be categorized into 5 categories:

1. **Counting**: Counting questions ask for the number of objects meeting specific criteria (*e.g. "How many other things are there of the same size as the tiny metal block?"*).

2. **Existence**: Existence questions ask if an object with certain properties is present in the image (*e.g. "Are there any other things that are the same color as the cylinder?"*).

3. **Integer Comparison**: Integer comparison questions ask about the relative sizes of two sets of objects (*e.g. "Are there fewer spheres than large cylinders?"*).

4. **Comparing Attributes**: Attribute comparison questions check if two objects have the same value for an attribute (*e.g. "Do the red cube and the green cylinder have the same size?"*).

5. **Querying Attributes**: Query questions ask about an attribute of a particular object (*e.g. "What shape is the object at the right of the green cube?"*).

The set of possible answers includes yes, no, numerical values between 0 and the maximum number of objects in the dataset, and all possible values of object properties. We have a different number of templates per each question category in our question-generation process. The proportion of each question category in each dataset is outlined in Fig. 9.

For CLEVR, questions are generated based on the attributes of each object, including shape, color, size, material, and their relative position in comparison to other objects. In the case of Multi-dSprites, where objects lack a material attribute, questions focus on the shape, color, size, and relative position of the objects. Regarding CLEVRTex, questions are generated considering only shape, size, and relative position, as color is absent in the dataset and is replaced with 60 materials, each having a specific color. The material is not included as a feature for the questions because the names of the materials are not suitable for accurate processing by a text embedding module.

**Real-World.** The VQA-v2 dataset includes questions from three categories: *Yes/No*, *Number*, and *Other*, with open-ended answers that are not limited to specific words or numbers. Therefore, in order to use the same framework employed for synthetic datasets, we only keep a subset of questions with specific answers. We omit questions with "Other" answer types and keep "Yes/No" questions that have "yes" or "no" answers. Additionally, we retain questions with numerical answers ranging from 0 to 14, resulting in a total of 17 possible answers in the "Yes/No" and "Number" categories. Before filtering, the train and test sets contain 443,757 and 214,354 questions, respectively. After filtering, they are reduced to 215,553 and 103,717.

The GQA dataset contains questions from five structural categories: *Verify*, *Logical*, *Compare*, *Query*, *Choose*. *Verify* questions are yes/no queries that check for the presence or attributes of objects. *Logical* questions demand inference and multi-step reasoning. *Compare* questions require evaluating two or more objects against each other. *Query* questions are open-ended, asking for descriptive details

about the scene. Finally, *Choose* questions present two alternatives from which the answer must be selected. Furthermore, unlike VQA-v2, on GQA, the number of unique answers is not significantly large, making it possible to use the pipeline as-is without any deletion.

## C   METRICS

**Upstream.**   To evaluate the performance of upstream models trained on the datasets used in our study, we use MSE reconstruction error and Adjusted Rand Index (ARI) (Hubert & Arabie, 1985), which are two commonly used metrics in the literature (Locatello et al., 2020; Dittadi et al., 2022; Singh et al., 2022b; Biza et al., 2023). Consistent with prior work, we calculate the ARI considering only foreground objects. Additionally, following Dittadi et al. (2022), we include Segmentation Covering (Arbelaez et al., 2010) and mean Segmentation Covering (Engelcke et al., 2019) as evaluation metrics. For more information on the upstream metrics, we refer to Dittadi et al. (2022).

**Downstream VQA.**   To assess the performance of representations in the VQA downstream task, following previous works on applying the VQA task to object-centric models (Ding et al., 2021a; Wu et al., 2022; Johnson et al., 2017) and other works on VQA in general (Antol et al., 2015; Goyal et al., 2017; Ren et al., 2015; Yang et al., 2022), we use accuracy as our main metric. We also analyzed balanced accuracy which takes into account the class imbalances and is defined as the average of recall obtained in each class, and F1 score as alternative metrics but our results revealed consistent trends across all metrics. Consequently, we focus on presenting results based on the accuracy metric.

## D   ADDITIONAL RESULTS

In this section, we report additional results that did not fit into the main part.

### D.1   PROPERTY PREDICTION VS VQA

Fig. 10 shows Spearman's rank correlation between downstream property prediction performance of the models for each property, and downstream VQA performance for each question category. Overall, we observe a strong correlation between the performance of the models in two downstream tasks. It's important to note that training for property prediction takes significantly less time than VQA, by around 2 orders of magnitude. Hence, property prediction can be a helpful guide when selecting a model for a downstream task.

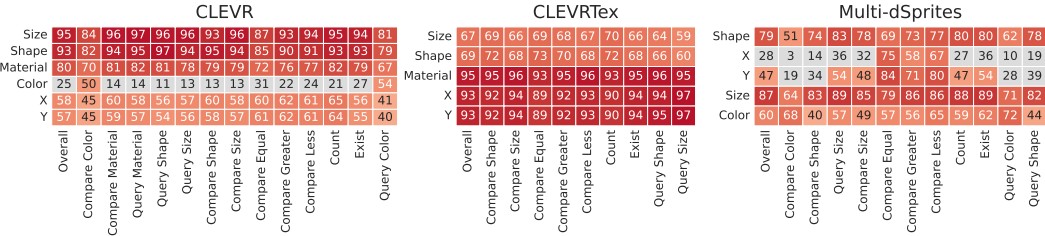

Figure 10: Spearman's rank correlation between downstream property prediction performance and downstream VQA performance of the models. The correlations are color-coded only when $p < 0.05$.

### D.2   UPSTREAM VS DOWNSTREAM PERFORMANCE

Here we analyze the relationship between the upstream and downstream VQA performance of OC models. We exclude VQA-v2 and GQA results due to only having three well-performing OC models for this dataset. Figs. 11 to 13 depict the upstream performance of object-centric models in comparison to their downstream performance when using T-15 as the downstream model. Generally, there is no strong correlation between the two performances, and several outliers are observed in each plot.

More specifically, on CLEVR and Multi-dSprites, only ARI on CLEVR shows a slight correlation with VQA accuracy. All other upstream metrics do not correlate with the downstream performance. On CLEVRTex, however, we observe a weak correlation between ARI, mSC, and SC with VQA accuracy. At the same time, contrary to expectations, we observe a positive correlation between MSE and overall VQA accuracy which suggests that models with higher MSE values tend to perform better on the downstream task. These require further investigation which is beyond the scope of this work.

Additionally, when we look at the outliers, STEVE is the main one among OC models in all datasets, showing higher MSE and lower ARI, but better downstream performance. Also, one seed of STEVE tends to perform poorly in upstream metrics but achieves a downstream performance comparable to other seeds. Moreover, DINOSAURv2 consistently performs poorly on mSC and SC across all datasets despite performing well downstream. Additionally, SPACE typically has the best MSE but the worst downstream performance.

In conclusion, we observe that upstream metrics are not a good indicator of the downstream performance of different models and thus, are not reliable for upstream model selection.

It is noteworthy that DINOSAURv2 does not reconstruct the input but instead reconstructs the latent features of the input. Thus, it is not included in the plots showing reconstruction MSE. Furthermore, we also calculated the Spearman rank correlations between upstream and downstream metrics but due to the high p-value of most of the correlations, we chose not to report them.

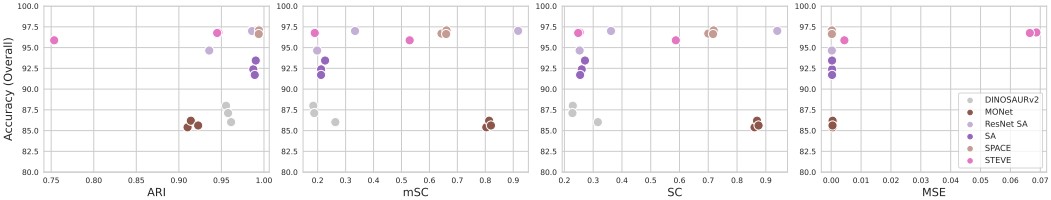

Figure 11: Upstream performance of object-centric models against the overall VQA accuracy when using T-15 as the downstream model on CLEVR.

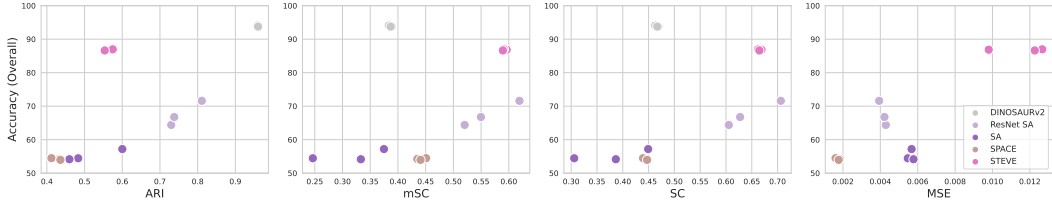

Figure 12: Upstream performance of object-centric models against the overall VQA accuracy when using T-15 as the downstream model on CLEVRTex.

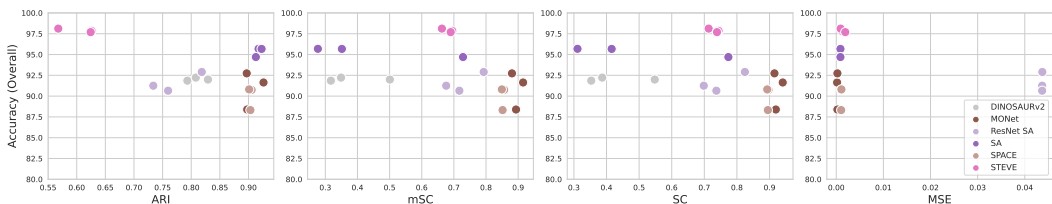

Figure 13: Upstream performance of object-centric models against the overall VQA accuracy when using T-15 as the downstream model on Multi-dSprites.

### D.3 EFFECT OF DOWNSTREAM MODEL SIZE

Fig. 14 depicts the overall accuracy of the models across different datasets with a downstream model having a varying number of layers. As expected, the downstream performance increases with the downstream model size. Furthermore, we observe that DINOSAURv2 performs better than DINOv2 on CLEVRTex, Multi-dSprites, and GQA when using T-2 as the downstream model. However, as the downstream model size increases, DINOv2's performance matches DINOSAURv2. This indicates that while DINOv2 representations do have the necessary information for object-related tasks, the information is less obvious and needs a larger downstream model to be effectively extracted. However, on CLEVR, we don't see the same pattern, as DINOSAURv2 performs slightly worse than DINOv2. After experimenting with various hyperparameter sets, we found the results of DINOSAURv2 on CLEVR to be highly sensitive to the selected hyperparameters, which likely explains its suboptimal performance.

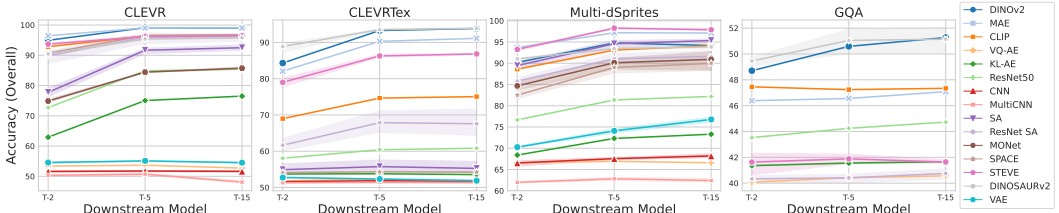

Figure 14: Average accuracies of different models w.r.t. downstream model size across different datasets. For pre-trained models, only one seed is available. For other models, the results are averaged over 3 random seeds and the shaded areas indicate 95% confidence intervals.

### D.4 EFFECT OF TRAINING SIZE

Fig. 15 depicts the overall accuracy of different models on Multi-dSprites with varying training sizes of 40k, 80k, 160k, and 320k unique images, and Fig. 16 shows the average percentage of decrease in error rate when increasing the dataset size from 40k to the respective size. With increased data, most upstream models show similar performance improvement across different training sizes regardless of their initial performance. CLIP is the only exception, showing a decrease in overall error rate of up to 50%. Furthermore, the performance of the end-to-end CNN and MultiCNN models show minimal improvement compared to the other models.

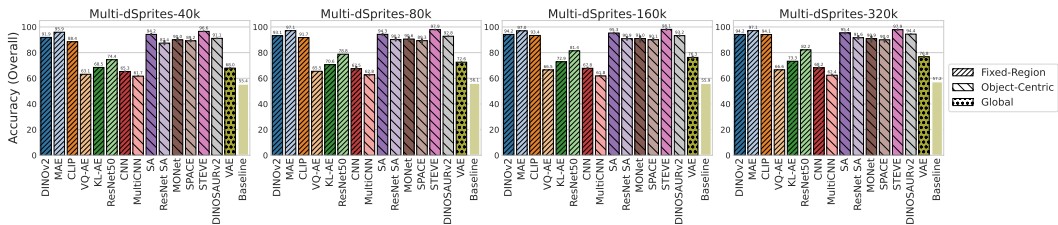

Figure 15: Average accuracies on the VQA downstream task for different models on Multi-dSprites with different training sizes when using T-15 as the downstream model. The bars indicate means and 95% confidence intervals with 3 random seeds.

### D.5 CONSISTENCY OF THE RESULTS ACROSS DIFFERENT QUESTION TYPES

Fig. 17 shows the Spearman's rank correlation between the performances of models on each question category when using T-15 for synthetic datasets and GQA, and T-2 for VQA-v2. Additionally, Figs. 18 to 28 illustrate the accuracy of different question types of all upstream model representations across different datasets. Except for GQA, we observe strong correlations between the performance of different question categories which indicates that the trend in the overall accuracy in Fig. 2 matches the trend in the accuracy of each question category and the results are consistent across different question categories.

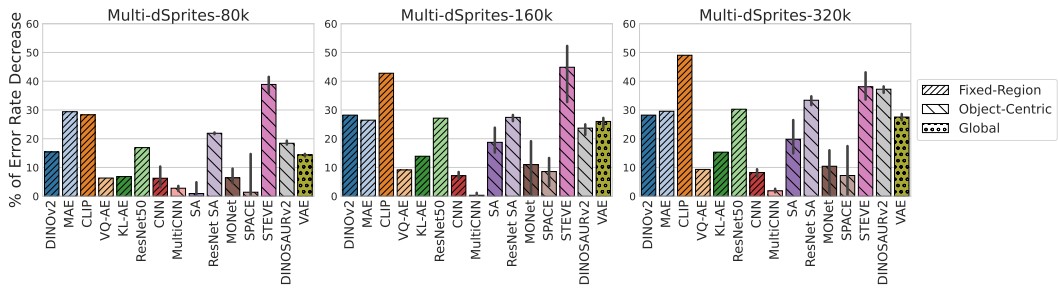

Figure 16: Average % decrease in VQA error rate for different upstream models on Multi-dSprites, when increasing the training size from 40k to larger sizes, using T-15 as the downstream model. The bars indicate means and 95% confidence intervals with 3 random seeds.

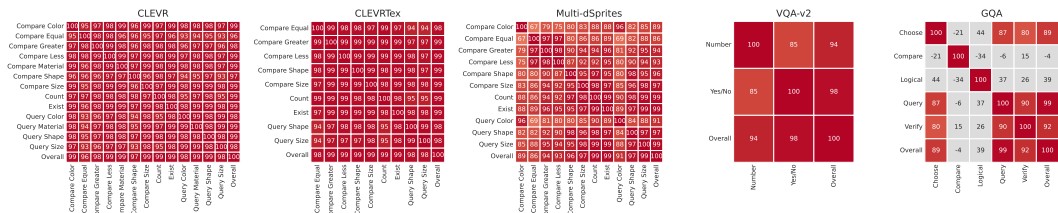

Figure 17: Spearman's rank correlation of model performances for each question category using T-15 as the downstream model.

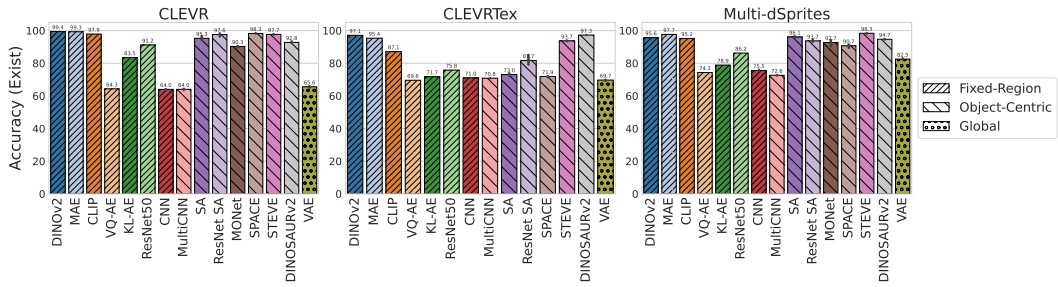

Figure 18: Average accuracies of *Exist* questions for different upstream representation models when using T-15 as the downstream model. The bars indicate means and 95% confidence intervals with 3 random seeds.

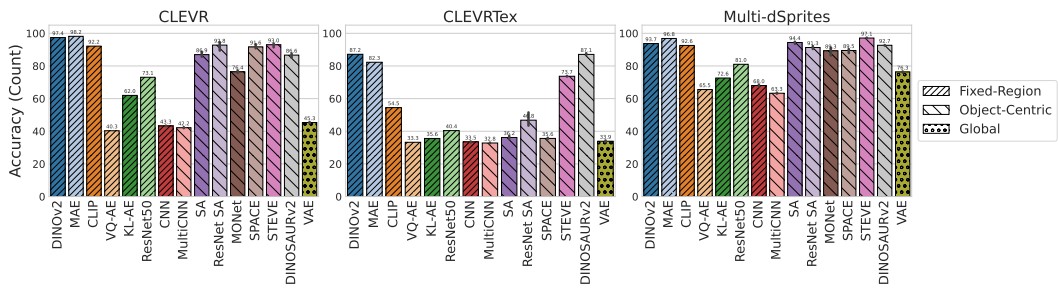

Figure 19: Average accuracies of *Count* questions for different upstream representation models when using T-15 as the downstream model. The bars indicate means and 95% confidence intervals with 3 random seeds.

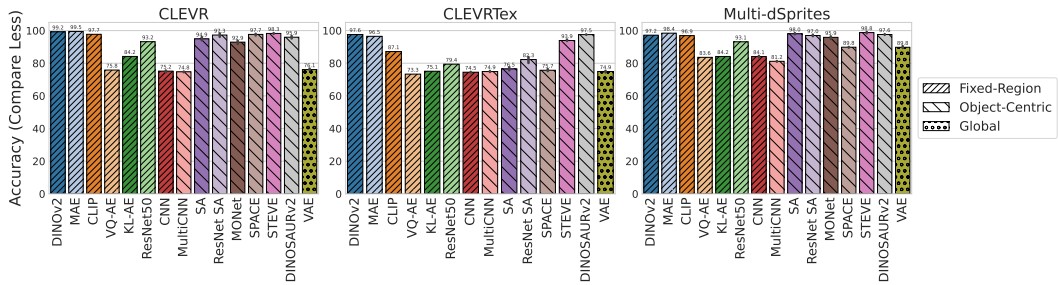

Figure 20: Average accuracies of *Compare Integer (Less)* questions for different upstream representation models when using T-15 as the downstream model. The bars indicate means and 95% confidence intervals with 3 random seeds.

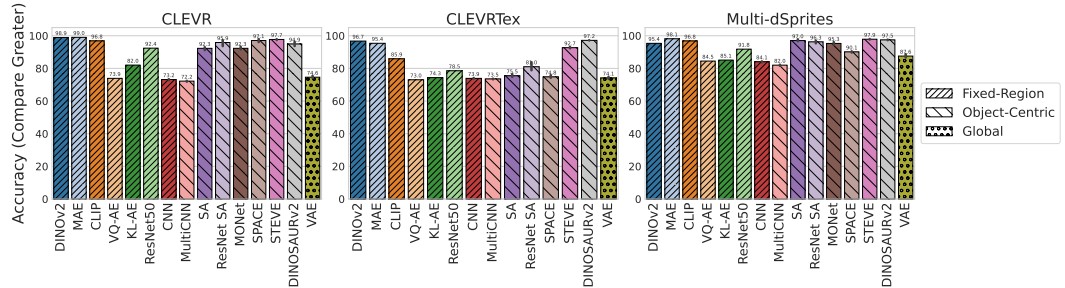

Figure 21: Average accuracies of *Compare Integer (Greater)* questions for different upstream representation models when using T-15 as the downstream model. The bars indicate means and 95% confidence intervals with 3 random seeds.

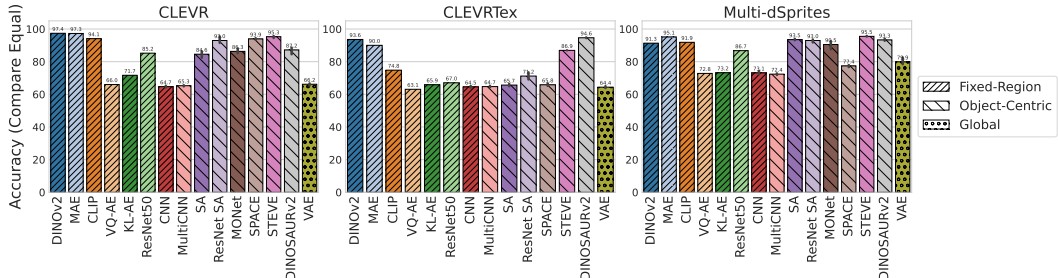

Figure 22: Average accuracies of *Compare Integer (Equal)* questions for different upstream representation models when using T-15 as the downstream model. The bars indicate means and 95% confidence intervals with 3 random seeds.

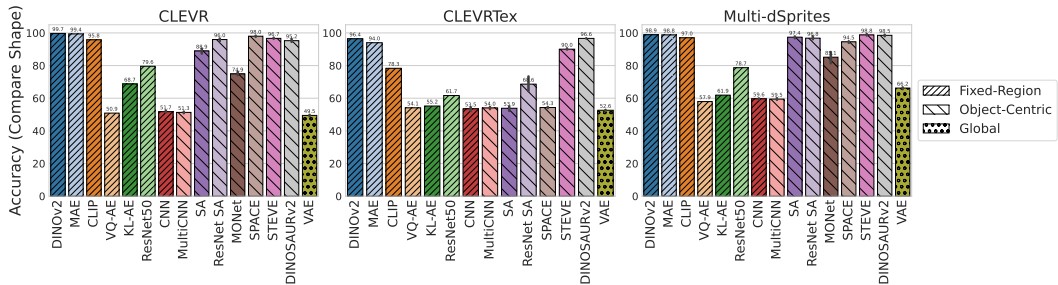

Figure 23: Average accuracies of *Compare Attribute (Shape)* questions for different upstream representation models when using T-15 as the downstream model. The bars indicate means and 95% confidence intervals with 3 random seeds.

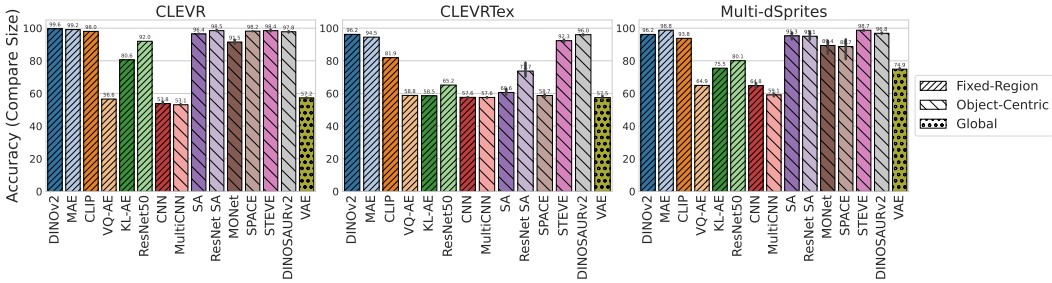

Figure 24: Average accuracies of *Compare Attribute (Size)* questions for different upstream representation models when using T-15 as the downstream model. The bars indicate means and 95% confidence intervals with 3 random seeds.

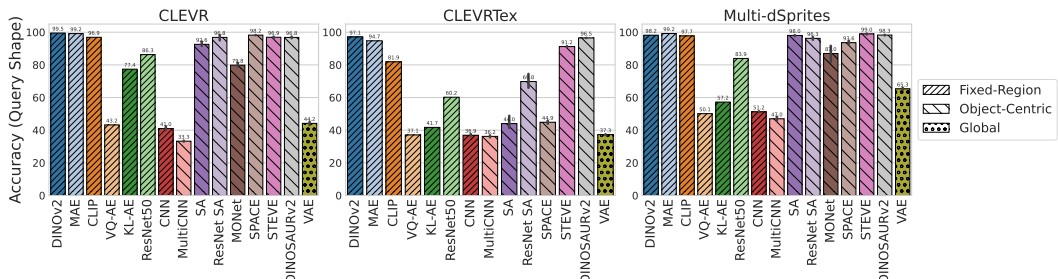

Figure 25: Average accuracies of *Query Attribute (Shape)* questions for different upstream representation models when using T-15 as the downstream model. The bars indicate means and 95% confidence intervals with 3 random seeds.

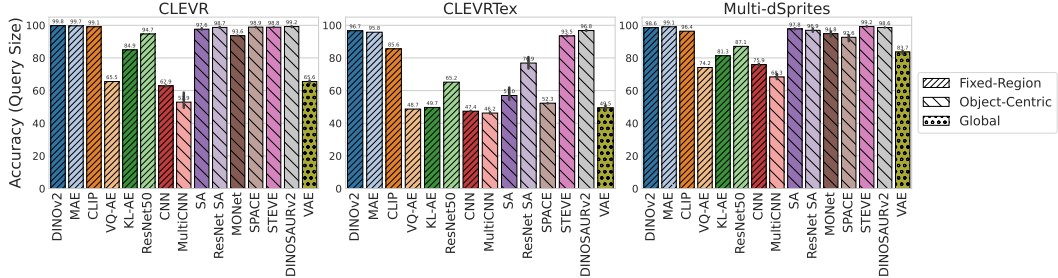

Figure 26: Average accuracies of *Query Attribute (Size)* questions for different upstream representation models when using T-15 as the downstream model. The bars indicate means and 95% confidence intervals with 3 random seeds.

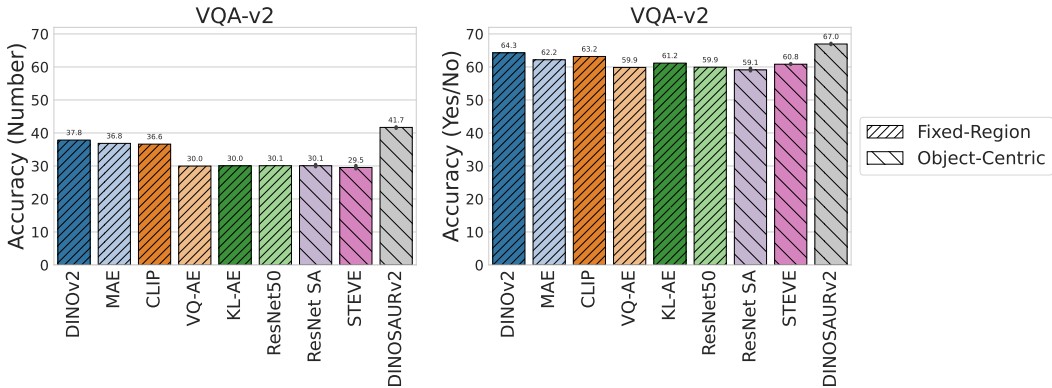

Figure 27: Average accuracies of different question types for different upstream representation models when using T-2 as the downstream model on VQA-v2. The bars indicate means and 95% confidence intervals with 3 random seeds.

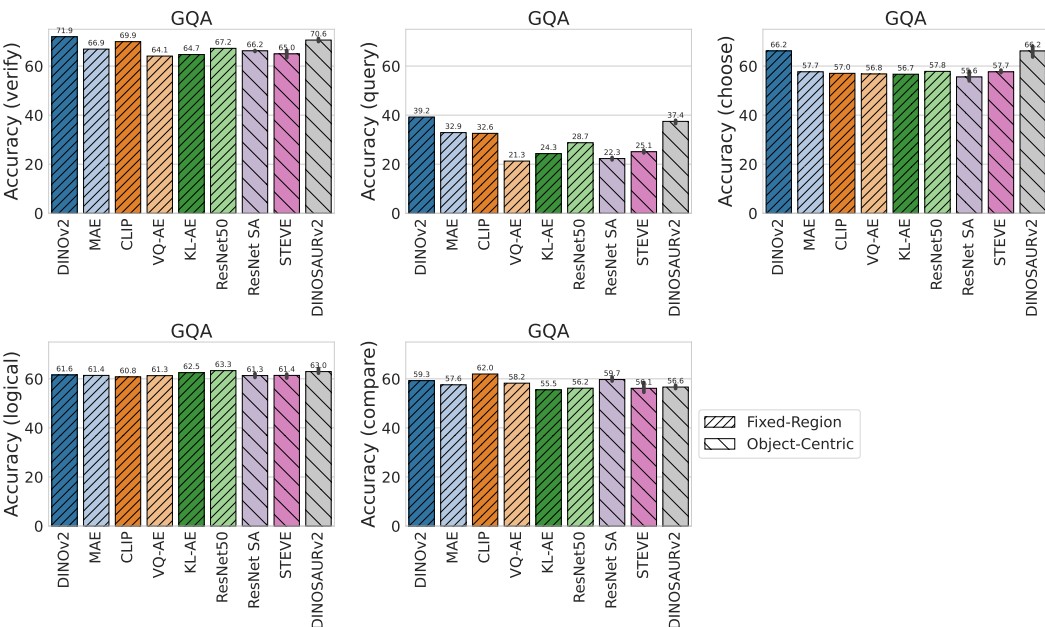

Figure 28: Average accuracies of different question types for different upstream representation models when using T-15 as the downstream model on GQA. The bars indicate means and 95% confidence intervals with 3 random seeds.

### D.6 FULL RESULTS

The complete VQA accuracies of different upstream models across all datasets are presented in Tables 7 to 14.

Table 7: Average accuracies on CLEVR when using T-15 as the downstream model. For pre-trained models, only one seed is available. For other models, the results are aggregated over 3 random seeds.

| Model | Overall | Exist | Count | Compare Integer | | | Compare Attribute | | | | Query Attribute | | | |
|---|---|---|---|---|---|---|---|---|---|---|---|---|---|---|
| | | | | Less | Greater | Equal | Shape | Color | Material | Size | Shape | Color | Material | Size |
| DINOv2 | 98.9 | 99.4 | 97.4 | 99.2 | 98.9 | 97.4 | 99.7 | 100.0 | 97.4 | 99.6 | 99.5 | 99.7 | 99.3 | 99.8 |
| MAE | 99.0 | 99.3 | 98.2 | 99.5 | 99.0 | 97.3 | 99.4 | 99.9 | 98.2 | 99.2 | 99.2 | 99.6 | 99.5 | 99.7 |
| CLIP | 96.5 | 97.9 | 92.2 | 97.7 | 96.8 | 94.1 | 95.8 | 98.6 | 96.2 | 98.0 | 96.9 | 98.2 | 98.1 | 99.1 |
| VQ-AE | 52.7 | 64.3 | 40.3 | 75.8 | 73.9 | 66.0 | 50.9 | 52.1 | 49.9 | 56.6 | 43.2 | 33.2 | 58.5 | 65.5 |
| KL-AE | 76.5 | 83.5 | 62.0 | 84.2 | 82.0 | 71.7 | 68.7 | 80.0 | 75.6 | 80.6 | 77.4 | 77.4 | 84.6 | 84.9 |
| ResNet50 | 85.4 | 91.2 | 73.1 | 93.2 | 92.4 | 85.2 | 79.6 | 85.1 | 82.3 | 92.0 | 86.3 | 84.9 | 89.6 | 94.7 |
| CNN | 51.6 | 64.0 | 43.3 | 75.2 | 73.2 | 64.7 | 51.7 | 53.2 | 51.3 | 53.8 | 41.0 | 21.1 | 59.0 | 62.9 |
| MultiCNN | 48.1 | 64.0 | 42.2 | 74.8 | 72.2 | 65.3 | 51.3 | 52.7 | 51.4 | 53.1 | 33.3 | 12.6 | 50.3 | 52.9 |
| SA | 92.5 | 95.3 | 86.9 | 94.9 | 92.3 | 84.6 | 88.9 | 93.7 | 92.7 | 96.4 | 92.6 | 93.5 | 95.8 | 97.6 |
| ResNet SA | 96.2 | 97.6 | 92.8 | 97.3 | 95.9 | 93.0 | 96.0 | 96.4 | 96.3 | 98.5 | 96.8 | 96.4 | 98.0 | 98.7 |
| MONet | 85.7 | 90.3 | 76.4 | 92.9 | 92.3 | 86.3 | 74.9 | 88.0 | 87.9 | 91.5 | 79.8 | 86.1 | 92.2 | 93.6 |
| SPACE | 96.8 | 98.3 | 91.6 | 97.7 | 97.1 | 93.9 | 98.0 | 98.0 | 97.5 | 98.2 | 98.2 | 98.5 | 98.8 | 98.9 |
| STEVE | 96.5 | 97.7 | 93.0 | 98.3 | 97.7 | 95.3 | 96.7 | 96.9 | 96.2 | 98.4 | 96.9 | 96.5 | 97.7 | 98.8 |
| DINOSAURv2 | 94.3 | 92.8 | 86.6 | 95.9 | 94.9 | 87.2 | 95.2 | 76.3 | 95.1 | 97.8 | 96.8 | 63.0 | 97.7 | 99.2 |
| VAE | 54.5 | 65.6 | 45.3 | 76.1 | 74.6 | 66.2 | 49.5 | 52.2 | 52.6 | 57.2 | 44.2 | 32.6 | 63.2 | 65.6 |

Table 8: Average accuracies on CLEVRTex when using T-15 as the downstream model. For pre-trained models, only one seed is available. For other models, the results are aggregated over 3 random seeds.

| Model | Overall | Exist | Count | Compare Integer | | | Compare Attribute | | Query Attribute | |
|---|---|---|---|---|---|---|---|---|---|---|
| | | | | Less | Greater | Equal | Shape | Size | Shape | Size |
| DINOv2 | 93.8 | 97.1 | 87.2 | 97.6 | 96.7 | 93.6 | 96.4 | 96.2 | 97.1 | 96.7 |
| MAE | 91.1 | 95.4 | 82.3 | 96.5 | 95.4 | 90.0 | 94.0 | 94.5 | 94.7 | 95.8 |
| CLIP | 75.1 | 87.1 | 54.5 | 87.1 | 85.9 | 74.8 | 78.3 | 81.9 | 81.9 | 85.6 |
| VQ-AE | 51.5 | 69.6 | 33.3 | 73.3 | 73.0 | 63.1 | 54.1 | 58.8 | 37.1 | 48.7 |
| KL-AE | 53.5 | 71.7 | 35.6 | 75.1 | 74.3 | 65.9 | 55.2 | 58.5 | 41.7 | 49.7 |
| ResNet50 | 60.8 | 75.8 | 40.4 | 79.4 | 78.5 | 67.0 | 61.7 | 65.2 | 60.2 | 65.2 |
| CNN | 51.7 | 71.0 | 33.5 | 74.5 | 73.9 | 64.5 | 53.5 | 57.6 | 36.9 | 47.4 |
| MultiCNN | 51.3 | 70.8 | 32.8 | 74.9 | 73.5 | 64.7 | 54.0 | 57.6 | 36.2 | 46.2 |
| SA | 55.3 | 73.0 | 36.2 | 76.5 | 75.5 | 65.7 | 53.9 | 60.6 | 44.0 | 57.0 |
| ResNet SA | 67.6 | 81.7 | 46.8 | 82.3 | 81.0 | 71.2 | 68.6 | 73.7 | 69.8 | 76.9 |
| SPACE | 54.2 | 71.9 | 35.6 | 75.7 | 74.8 | 65.8 | 54.3 | 58.7 | 44.9 | 52.3 |
| STEVE | 86.8 | 93.7 | 73.7 | 93.9 | 92.7 | 86.9 | 90.0 | 92.3 | 91.2 | 93.5 |
| DINOSAURv2 | 93.9 | 97.3 | 87.1 | 97.5 | 97.2 | 94.6 | 96.6 | 96.0 | 96.5 | 96.8 |
| VAE | 51.9 | 69.7 | 33.9 | 74.9 | 74.1 | 64.4 | 52.6 | 57.5 | 37.3 | 49.5 |

Table 9: Average accuracies on Multi-dSprites with 40k unique training images when using T-15 as the downstream model. For pre-trained models, only one seed is available. For other models, the results are aggregated over 3 random seeds.

| Model | Overall | Exist | Count | Compare Integer | | | Compare Attribute | | | Query Attribute | | |
|-------|---------|-------|-------|------|---------|-------|-------|-------|------|-------|-------|------|
| | | | | Less | Greater | Equal | Shape | Color | Size | Shape | Color | Size |
| DINOv2 | 91.9 | 94.1 | 91.3 | 96.4 | 96.1 | 91.7 | 96.9 | 80.8 | 94.1 | 97.6 | 77.4 | 96.1 |
| MAE | 95.9 | 96.9 | 95.2 | 98.1 | 98.2 | 94.8 | 98.2 | 91.0 | 96.7 | 98.8 | 89.9 | 98.2 |
| CLIP | 88.4 | 91.0 | 86.6 | 92.8 | 93.1 | 86.1 | 92.3 | 77.2 | 84.8 | 95.0 | 80.8 | 92.1 |
| VQ-AE | 63.1 | 72.8 | 62.2 | 82.6 | 83.2 | 71.8 | 53.5 | 58.8 | 59.9 | 48.6 | 41.6 | 71.5 |
| KL-AE | 68.5 | 75.4 | 67.9 | 83.2 | 84.2 | 72.1 | 56.9 | 66.4 | 65.0 | 54.2 | 56.8 | 76.5 |
| ResNet50 | 74.4 | 79.9 | 73.7 | 87.8 | 87.3 | 78.8 | 73.4 | 60.9 | 70.0 | 77.4 | 53.5 | 79.8 |
| CNN | 65.3 | 74.0 | 65.9 | 82.3 | 82.6 | 71.6 | 56.3 | 62.1 | 59.9 | 49.8 | 47.4 | 72.5 |
| MultiCNN | 61.7 | 72.6 | 62.6 | 80.7 | 80.7 | 71.3 | 58.3 | 59.0 | 58.0 | 46.6 | 32.2 | 68.4 |
| SA | 94.2 | 95.4 | 93.1 | 97.2 | 96.4 | 92.2 | 96.2 | 87.8 | 94.7 | 97.4 | 89.2 | 97.2 |
| ResNet SA | 87.4 | 90.6 | 87.2 | 95.5 | 93.6 | 89.0 | 94.4 | 65.0 | 90.4 | 94.7 | 65.3 | 95.7 |
| MONet | 90.0 | 92.4 | 88.2 | 95.6 | 93.9 | 88.5 | 84.1 | 88.6 | 88.0 | 86.4 | 89.8 | 94.6 |
| SPACE | 89.2 | 90.3 | 88.8 | 89.3 | 89.2 | 76.3 | 93.5 | 85.9 | 89.8 | 93.2 | 85.8 | 92.0 |
| STEVE | 96.6 | 97.5 | 95.5 | 98.0 | 97.1 | 93.5 | 97.6 | 92.6 | 97.9 | 98.5 | 95.1 | 98.8 |
| DINOSAURv2 | 91.1 | 91.0 | 87.8 | 95.3 | 94.2 | 89.4 | 94.3 | 64.8 | 92.8 | 95.8 | 63.3 | 95.9 |
| VAE | 68.0 | 74.5 | 68.9 | 83.8 | 83.3 | 71.0 | 59.3 | 60.9 | 68.7 | 57.7 | 47.7 | 75.6 |

Table 10: Average accuracies on Multi-dSprites with 80k unique training images when using T-15 as the downstream model. For pre-trained models, only one seed is available. For other models, the results are aggregated over 3 random seeds.

| Model | Overall | Exist | Count | Compare Integer | | | Compare Attribute | | | Query Attribute | | |
|-------|---------|-------|-------|------|---------|-------|-------|-------|------|-------|-------|------|
| | | | | Less | Greater | Equal | Shape | Color | Size | Shape | Color | Size |
| DINOv2 | 93.1 | 95.3 | 92.9 | 97.0 | 96.0 | 91.6 | 97.7 | 81.2 | 94.7 | 98.2 | 81.1 | 97.4 |
| MAE | 97.1 | 97.7 | 96.6 | 98.2 | 98.3 | 96.5 | 98.3 | 93.0 | 98.7 | 98.7 | 93.5 | 99.1 |
| CLIP | 91.7 | 94.3 | 90.5 | 95.2 | 95.6 | 89.5 | 95.3 | 84.0 | 89.0 | 95.8 | 83.9 | 95.2 |
| VQ-AE | 65.5 | 73.0 | 64.6 | 83.3 | 83.5 | 73.4 | 57.8 | 62.8 | 62.5 | 50.0 | 49.0 | 73.0 |
| KL-AE | 70.6 | 76.9 | 67.5 | 84.3 | 85.1 | 70.8 | 60.8 | 70.7 | 70.1 | 57.1 | 63.6 | 80.3 |
| ResNet50 | 78.8 | 82.7 | 77.6 | 90.6 | 89.4 | 83.5 | 76.5 | 66.3 | 74.5 | 81.7 | 62.5 | 85.0 |
| CNN | 67.5 | 75.3 | 67.4 | 83.4 | 83.7 | 72.8 | 58.9 | 65.1 | 63.5 | 51.2 | 53.0 | 75.1 |
| MultiCNN | 62.8 | 72.7 | 63.6 | 81.2 | 82.0 | 72.4 | 59.5 | 61.3 | 59.9 | 48.2 | 34.0 | 68.9 |
| SA | 94.3 | 95.4 | 93.6 | 97.8 | 97.0 | 93.4 | 97.1 | 89.0 | 91.8 | 97.7 | 88.2 | 96.0 |
| ResNet SA | 90.2 | 93.1 | 89.7 | 96.0 | 95.4 | 91.0 | 96.0 | 71.9 | 93.3 | 95.5 | 73.5 | 96.3 |
| MONet | 90.6 | 92.3 | 88.6 | 95.8 | 94.5 | 91.0 | 84.9 | 90.9 | 89.6 | 86.8 | 91.5 | 94.7 |
| SPACE | 89.3 | 90.1 | 88.3 | 89.6 | 89.2 | 77.3 | 93.9 | 89.1 | 87.4 | 93.3 | 87.6 | 91.9 |
| STEVE | 97.9 | 98.4 | 97.3 | 98.9 | 98.3 | 95.8 | 99.3 | 95.8 | 99.0 | 99.1 | 96.4 | 99.3 |
| DINOSAURv2 | 92.8 | 93.6 | 90.6 | 96.1 | 96.0 | 91.7 | 97.0 | 74.5 | 95.0 | 96.9 | 75.3 | 97.2 |
| VAE | 72.6 | 78.2 | 72.2 | 85.0 | 85.3 | 73.4 | 65.5 | 67.7 | 71.3 | 62.7 | 60.6 | 79.5 |

Table 11: Average accuracies on Multi-dSprites with 160k unique training images when using T-15 as the downstream model. For pre-trained models, only one seed is available. For other models, the results are aggregated over 3 random seeds.

| Model | Overall | Exist | Count | Compare Integer | | | Compare Attribute | | | Query Attribute | | |
|-------|---------|-------|-------|------|---------|-------|-------|-------|------|-------|-------|------|
| | | | | Less | Greater | Equal | Shape | Color | Size | Shape | Color | Size |
| DINOv2 | 94.2 | 95.6 | 93.7 | 97.2 | 95.4 | 91.3 | 98.9 | 84.9 | 96.2 | 98.2 | 84.8 | 98.6 |
| MAE | 97.0 | 97.6 | 96.7 | 98.4 | 98.8 | 96.0 | 98.3 | 92.7 | 98.7 | 99.2 | 91.9 | 98.9 |
| CLIP | 93.4 | 95.5 | 91.6 | 96.6 | 96.5 | 91.1 | 96.7 | 90.5 | 91.5 | 97.1 | 86.9 | 96.0 |
| VQ-AE | 66.5 | 73.2 | 65.6 | 82.5 | 84.2 | 72.5 | 58.1 | 66.6 | 64.8 | 50.1 | 52.5 | 74.2 |
| KL-AE | 72.9 | 78.4 | 71.0 | 84.8 | 85.5 | 74.0 | 61.5 | 72.8 | 72.9 | 58.2 | 67.3 | 81.7 |
| ResNet50 | 81.4 | 86.0 | 80.0 | 92.8 | 90.8 | 85.3 | 78.2 | 70.3 | 76.4 | 84.2 | 66.8 | 87.1 |
| CNN | 67.8 | 75.4 | 67.4 | 83.8 | 83.8 | 72.8 | 60.0 | 65.3 | 62.7 | 51.2 | 54.8 | 75.8 |
| MultiCNN | 61.8 | 72.6 | 63.1 | 81.2 | 81.7 | 72.5 | 59.0 | 61.0 | 58.9 | 42.8 | 33.2 | 67.9 |
| SA | 95.3 | 96.0 | 94.3 | 98.2 | 97.8 | 94.7 | 97.5 | 91.0 | 94.5 | 98.0 | 91.0 | 97.2 |
| ResNet SA | 90.9 | 93.1 | 90.6 | 96.7 | 96.0 | 91.3 | 95.8 | 75.6 | 92.9 | 96.0 | 75.6 | 96.2 |
| MONet | 91.0 | 92.8 | 89.3 | 95.9 | 95.3 | 91.3 | 85.6 | 91.4 | 88.6 | 87.4 | 91.3 | 94.7 |
| SPACE | 90.1 | 90.6 | 89.6 | 89.6 | 90.0 | 77.6 | 94.8 | 89.8 | 89.0 | 93.8 | 88.0 | 92.6 |
| STEVE | 98.1 | 98.4 | 97.7 | 98.7 | 98.6 | 95.9 | 98.8 | 96.1 | 99.0 | 99.3 | 96.8 | 99.4 |
| DINOSAURv2 | 93.2 | 93.3 | 90.4 | 96.2 | 95.7 | 91.5 | 96.5 | 73.2 | 95.2 | 96.8 | 76.0 | 97.1 |
| VAE | 76.3 | 81.9 | 75.6 | 89.0 | 87.8 | 79.4 | 68.0 | 70.9 | 74.6 | 65.3 | 67.2 | 83.2 |

Table 12: Average accuracies on Multi-dSprites with 320k unique training images when using T-15 as the downstream model. For pre-trained models, only one seed is available. For other models, the results are aggregated over 3 random seeds.

| Model | Overall | Exist | Count | Compare Integer | | | Compare Attribute | | | Query Attribute | | |
|---|---|---|---|---|---|---|---|---|---|---|---|---|
| | | | | Less | Greater | Equal | Shape | Color | Size | Shape | Color | Size |
| DINOv2 | 94.2 | 95.6 | 93.7 | 97.2 | 95.4 | 91.3 | 98.9 | 84.9 | 96.2 | 98.2 | 84.8 | 98.6 |
| MAE | 97.1 | 97.7 | 96.8 | 98.4 | 98.1 | 95.1 | 98.8 | 92.5 | 98.8 | 99.2 | 92.9 | 99.1 |
| CLIP | 94.1 | 95.2 | 92.6 | 96.9 | 96.8 | 91.9 | 97.0 | 90.3 | 93.8 | 97.7 | 89.3 | 96.4 |
| VQ-AE | 66.6 | 74.3 | 65.5 | 83.6 | 84.5 | 72.8 | 57.9 | 64.0 | 64.9 | 50.1 | 51.9 | 74.2 |
| KL-AE | 73.3 | 78.9 | 72.6 | 84.2 | 85.1 | 73.2 | 61.9 | 72.9 | 75.5 | 57.2 | 67.4 | 81.3 |
| ResNet50 | 82.2 | 86.2 | 81.0 | 93.1 | 91.8 | 86.7 | 78.7 | 71.1 | 80.1 | 83.9 | 68.0 | 87.1 |
| CNN | 68.2 | 75.5 | 68.0 | 84.1 | 84.1 | 73.1 | 59.6 | 66.2 | 64.8 | 51.2 | 55.0 | 75.9 |
| MultiCNN | 62.4 | 72.6 | 63.3 | 81.2 | 82.0 | 72.4 | 59.5 | 61.0 | 59.1 | 47.0 | 33.7 | 68.3 |
| SA | 95.4 | 96.1 | 94.4 | 98.0 | 97.0 | 93.5 | 97.4 | 90.4 | 95.3 | 98.0 | 91.2 | 97.8 |
| ResNet SA | 91.6 | 93.7 | 91.3 | 97.0 | 96.3 | 93.0 | 96.8 | 78.4 | 95.1 | 96.3 | 75.5 | 96.9 |
| MONet | 90.9 | 92.7 | 89.3 | 95.9 | 95.3 | 90.5 | 85.1 | 91.4 | 89.4 | 87.0 | 91.4 | 94.8 |
| SPACE | 90.0 | 90.7 | 89.5 | 89.8 | 90.1 | 77.4 | 94.5 | 88.7 | 88.7 | 93.6 | 87.4 | 92.6 |
| STEVE | 97.9 | 98.3 | 97.1 | 98.8 | 97.9 | 95.5 | 98.8 | 96.4 | 98.7 | 99.0 | 97.1 | 99.2 |
| DINOSAURv2 | 94.4 | 94.7 | 92.7 | 97.6 | 97.5 | 93.3 | 98.5 | 78.0 | 96.8 | 98.3 | 78.0 | 98.6 |
| VAE | 76.8 | 82.5 | 76.3 | 89.8 | 87.6 | 79.9 | 66.2 | 71.5 | 74.9 | 65.3 | 68.8 | 83.7 |

Table 13: Average accuracies on VQA-v2 when using T-2 as the downstream model. For pre-trained models, only one seed is available. For other models, the results are aggregated over 3 random seeds.

| Model | Overall | Number | Yes/No |
|---|---|---|---|
| DINOv2 | 58.4 | 37.8 | 64.3 |
| MAE | 56.5 | 36.8 | 62.2 |
| CLIP | 57.2 | 36.6 | 63.2 |
| VQ-AE | 53.2 | 30.0 | 59.9 |
| KL-AE | 54.2 | 30.0 | 61.2 |
| ResNet50 | 53.3 | 30.1 | 59.9 |
| ResNet SA | 52.6 | 30.1 | 59.1 |
| STEVE | 53.8 | 29.5 | 60.8 |
| DINOSAURv2 | 61.3 | 41.7 | 67.0 |

Table 14: Average accuracies on GQA when using T-15 as the downstream model. For pre-trained models, only one seed is available. For other models, the results are aggregated over 3 random seeds.

| Model | Overall | Verify | Query | Choose | Logical | Compare |
|---|---|---|---|---|---|---|
| DINOv2 | 51.3 | 71.9 | 39.2 | 66.2 | 61.6 | 59.3 |
| MAE | 47.1 | 66.9 | 32.9 | 57.7 | 61.4 | 57.6 |
| CLIP | 47.3 | 69.9 | 32.6 | 57.0 | 60.8 | 62.0 |
| VQ-AE | 40.6 | 64.1 | 21.3 | 56.8 | 61.3 | 58.2 |
| KL-AE | 41.7 | 64.7 | 24.3 | 56.7 | 62.5 | 55.5 |
| ResNet50 | 44.7 | 67.2 | 28.7 | 57.8 | 63.3 | 56.2 |
| ResNet SA | 40.8 | 66.2 | 22.3 | 55.6 | 61.3 | 59.7 |
| STEVE | 41.6 | 65.0 | 25.1 | 57.7 | 61.4 | 56.1 |
| DINOSAURv2 | 51.1 | 70.6 | 37.4 | 66.2 | 63.0 | 56.6 |

