# OpenReview forum: "Exploring the Effectiveness of Object-Centric Representations in Visual Question Answering: Comparative Insights with Foundation Models"
_ICLR.cc/2025/Conference — ICLR 2025 Poster_

### Official Review · Reviewer_feeJ · 2024-11-01

**Soundness:** 3
**Presentation:** 3
**Contribution:** 3
**Rating:** 6
**Confidence:** 3

**Summary:**

This paper explores the effectiveness of object-centric (OC) representations in the Visual Question Answering (VQA) task and compares them with traditional large-scale pretrained foundation models.

**Strengths:**

1. The experimental design is rigorous, considering the impact of multiple factors on the task.
2. The writing is fluent, and the conclusions are clear.
3. The finding that object-centric representations are effective in VQA tasks is insightful for the community.

**Weaknesses:**

1. As stated in the paper, the datasets would have included more real-world data.
2. Some analysis of the application of the findings is lacking.

**Questions:**

1. What makes me curious is what the performance would be after fine-tuning the foundation models on the specific VQA tasks, since other models have been fitted on the data. I suppose this would make it more fair for the comparison of different models and may inspire more insights.
2. Could you give some more insights on the application of the findings in the paper? I'm wondering if the conclusions would be applied in other domains out of VQA.

---

> ### Author Response · Authors · 2024-11-15
>
> Thank you for your review and your positive assessment of our work. We answer your questions and concerns below:
>
> ### Weaknesses and Questions
>
> > As stated in the paper, the datasets would have included more real-world data.
>
> We kindly refer you to our general response to all reviewers, where we explain that we are incorporating the GQA dataset as an additional real-world dataset into our study and will share the results as soon as they are ready.
>
> > Some analysis of the application of the findings is lacking.
>
> > Could you give some more insights on the application of the findings in the paper?
>
> We would like to highlight that the role of OC models in downstream tasks has been partially underexplored and our empirical study tries to cover this gap in the literature. We have summarised some of the takeaways of our study and their contribution to the community below:
>
> * We’ve empirically clarified the trade-offs of foundation models vs OC models and found out that foundation models will perform the same or better than standard OC models on complex or real-world data without any training or hyperparameter tuning. However, they require much more downstream compute and they have less explicit representations compared to OC models. However, applying the OC bias on top of a foundation model (DINOSAUR) will benefit us with the best of both worlds, leveraging the general vision capabilities of foundation models and thus, showing strong performance on hard synthetic and real-world datasets while having explicit object representations. The only downside is that the OC bias should be trained on the downstream dataset. Therefore, depending on the settings at hand, one can select the proper type of model based on the insights of our study.
>
> * For evaluation, the focus of the OC community has been mainly on unsupervised object discovery tasks and segmentation metrics while downstream evaluation of representations has often been overlooked. We’ve empirically shown that representations that perform well in downstream tasks do not necessarily excel in segmentation tasks, and that segmentation accuracy (ARI) or reconstruction error (MSE) are not reliable predictors of downstream performance. We hope this highlights the importance of incorporating downstream evaluation as a key aspect of model assessment in the OC community.
>
> > I'm wondering if the conclusions would be applied in other domains out of VQA.
>
> We would like to emphasize that VQA involves a diverse range of questions that require understanding and reasoning about the visual scene. Demonstrating that the methods work effectively on this task suggests that the results can be extended to other tasks that require similar levels of understanding and reasoning.
>
> > What makes me curious is what the performance would be after fine-tuning the foundation models on the specific VQA tasks, since other models have been fitted on the data. I suppose this would make it more fair for the comparison of different models and may inspire more insights.
>
> That’s a good point. In our initial set of experiments, we analyzed the effect of fine-tuning DINOv2 on the VQA task, and as expected, we observed that the downstream performance improved but the gain was not significant and came with a substantial computational burden. This made the comparison with OC models less fair in terms of computational requirements. Thus, we decided not to pursue fine-tuning further.

---

> > ### Author Response · Authors · 2024-11-21
> >
> > Dear reviewer, thank you once again for taking the time to review our submission. We’d appreciate it if you could let us know whether the additional experiments and our responses address your concerns and if you might consider adjusting your score accordingly. We’re also open to any further feedback or questions you may have during the discussion period.

---

> > > ### Author Response · Authors · 2024-11-25
> > >
> > > Dear Reviewer,
> > >
> > > Thank you for the time and effort you have dedicated to reviewing our work. Your thoughtful feedback has been instrumental in enhancing the clarity and rigor of our paper.
> > >
> > > As the discussion period is coming to a close, we wanted to ensure that our responses along with the additional experiments on the GQA dataset have addressed your concerns. If we have successfully addressed your points, we would be grateful if you could kindly consider increasing your score. Of course, we remain happy to address any additional points if needed.
> > >
> > > Thank you once again for your valuable feedback.

---

> ### Comment · Reviewer_feeJ · 2024-11-25
>
> I thank the authors for their efforts and their response has addressed my concerns to some extent. Hence, I decide to keep my original score and am leaning to accept this paper.

---

> > ### Author Response · Authors · 2024-11-25
> >
> > Thank you for acknowledging our rebuttal and confirming your positive assessment of our work. We sincerely appreciate the time and effort you dedicated to reviewing our paper and providing valuable feedback.

---

### Official Review · Reviewer_HU9k · 2024-11-03

**Soundness:** 2
**Presentation:** 4
**Contribution:** 2
**Rating:** 6
**Confidence:** 5

**Summary:**

This is an experimental paper that examines how object-centric (OC) representations, which treat scenes as compositions of objects, compare with large pre-trained foundation models in Visual Question Answering (VQA) tasks.

**Strengths:**

Using object features to represent images to solve VQA has become quite popular since the bottom-up attention (from Peter Anderson et al.). However, there have been many debates about whether the object-level feature makes it work or whether it is just because the image representation model used in BUTD was better trained (with more data). This paper tried to solve this problem by running many experiments.

**Weaknesses:**

1. The question that this paper wants to answer is: whether OC, fixed-region or global representation is better for VQA. To answer this question, the authors need to provide a model with the same architecture, model size and pre-training data. The only variant in the model is the feature representation way, i.e., OC, fixed-region or global. However, I didn't find this experiment in the paper. Maybe I missed it. Please point it out and highlight it in the rebuttal.

2. Synthetic data such as CLEVR and CLEVRTex are not challenging enough since the objects can be easily detected (and even hard-coded). Many symbolic methods have solved these datasets with 100% accuracy.

3. I am not sure how these takeaway messages could contribute to the community. Some conclusions are already known such as : ' Consistency of the Results Across Question Types.'

**Questions:**

See weakness, I would like to see the responses to the above three weaknesses.

---

> ### Author Response · Authors · 2024-11-15
>
> Thank you for your review of our work and the detailed feedback. We answer your questions and concerns below:
>
> ### Weaknesses
>
> > The question that this paper wants to answer is: whether OC, fixed-region or global representation is better for VQA. To answer this question, the authors need to provide a model with the same architecture, model size and pre-training data. The only variant in the model is the feature representation way, i.e., OC, fixed-region or global. However, I didn't find this experiment in the paper. Maybe I missed it. Please point it out and highlight it in the rebuttal.
>
> Thank you for pointing this out. We completely agree that the fairest way to compare these different representations would be to maintain the same factors of variation such as model architecture, size, and pre-training data while varying only the representation type (OC, fixed-region, or global). However, implementing such a controlled experiment would require substantial engineering effort and computational resources to ensure all factors except the representation type remain consistent, and everything works as expected. In our study, we instead took a best-effort approach by utilizing existing models with minimal modifications, which allowed us to assess the effectiveness of different representations without setting up the entire experiment from scratch. While we acknowledge this as a limitation, we have made efforts to minimize the impact of other factors, particularly by comparing a foundation model with and without the OC inductive bias (DINOv2 and DINOSAURv2), ensuring as fair a comparison as possible within our available resources.
>
> > Synthetic data such as CLEVR and CLEVRTex are not challenging enough since the objects can be easily detected (and even hard-coded).
>
> We agree that solely using these synthetic datasets might not be enough for our study and therefore, we have included VQA-v2 to confirm our results in real-world settings. We have selected these datasets mainly because they are some of the traditional synthetic datasets that have been popular in the OC community in the past years.
> Furthermore, we kindly refer you to our general response to all reviewers, where we explain that we are incorporating the GQA dataset as an additional real-world dataset into our study and will share the results as soon as they are ready.
>
> > Many symbolic methods have solved these datasets with 100% accuracy.
>
> Can you please elaborate on which methods have achieved the perfect performance, and on which tasks?
>
> Nonetheless, we would like to highlight that this is not directly relevant to our study. The fact that some models can achieve perfect accuracy indicates that those models are effective at solving all the specific challenges presented by the dataset. However, for our study, the objective is not to solely push the boundaries of performance but to investigate how different model representations, including OC representations, behave on the VQA tasks defined on these datasets. In fact, our goal is not to solve the VQA task and achieve state-of-the-art results but to analyze the benefits and trade-offs of OC representations compared to other types of representations, with VQA serving as the downstream task for this evaluation. Therefore, as long as we observe a difference in the performance of models on these datasets and not all of the models achieve perfect accuracy, they would be suitable datasets for this goal.

---

> > ### Author Response · Authors · 2024-11-15
> >
> > > I am not sure how these takeaway messages could contribute to the community.
> >
> > We would like to highlight that the role of OC models in downstream tasks has been partially underexplored and our empirical study tries to cover this gap in the literature. We have summarised some of the takeaways of our study and their contribution to the community below:
> >
> > * We’ve empirically clarified the trade-offs of foundation models vs OC models and found out that foundation models will perform the same or better than standard OC models on complex or real-world data without any training or hyperparameter tuning. However, they require much more downstream compute and they have less explicit representations compared to OC models. However, applying the OC bias on top of a foundation model (DINOSAUR) will benefit us with the best of both worlds, leveraging general vision capabilities of foundation models and thus, showing strong performance on hard synthetic and real-world datasets while having explicit object representations. The main downside is that the OC bias should be trained on the downstream dataset. Therefore, depending on the settings at hand, one can select the proper type of model based on these insights.
> >
> > * For evaluation, the focus of the OC community has been mainly on unsupervised object discovery tasks and segmentation metrics while downstream evaluation of representations has often been overlooked. We’ve empirically shown that representations that perform well in downstream tasks do not necessarily excel in segmentation tasks, and that segmentation accuracy (ARI) or reconstruction error (MSE) are not reliable predictors of downstream performance. We hope this highlights the importance of incorporating downstream evaluation as a key aspect of model assessment in the OC community.
> >
> > > Some conclusions are already known such as : ' Consistency of the Results Across Question Types.'
> >
> > To the best of our knowledge, this conclusion has not been definitively demonstrated for OC models. We agree that the finding might not be surprising, but it’s not entirely obvious either. Specifically, given that the representations in our study are learned in very different ways, it could also be plausible that certain representations would perform better on specific question types, while others might excel on different types. This possibility makes the consistency of results across question types relevant. Even though it’s a relatively minor insight, we felt it was important to include, as we believe it adds value to the overall understanding and provides helpful documentation.
> >
> > We are happy to answer any further questions you may have.

---

> > > ### Author Response · Authors · 2024-11-21
> > >
> > > Dear reviewer, thank you once again for taking the time to review our submission. We’d appreciate it if you could let us know whether the additional experiments and our responses address your concerns and if you might consider adjusting your score accordingly. We’re also open to any further feedback or questions you may have during the discussion period.

---

> > > > ### Author Response · Authors · 2024-11-25
> > > >
> > > > Dear Reviewer,
> > > >
> > > > Thank you for the time and effort you have dedicated to reviewing our work. Your thoughtful feedback has been instrumental in enhancing the clarity and rigor of our paper.
> > > >
> > > > As the discussion period is coming to a close, we wanted to ensure that our responses along with the additional experiments on the GQA dataset have addressed your concerns. If we have successfully addressed your points, we would be grateful if you could kindly consider increasing your score. Of course, we remain happy to address any additional points if needed.
> > > >
> > > > Thank you once again for your valuable feedback.

---

> > > ### Comment · Reviewer_HU9k · 2024-11-25
> > >
> > > According to the findings and takeaway messages, I am sure they are correct (based on the experimental results you provided), but I agree with 7see that some (or most) conclusions are intuitive and well-known, which minimize their contribution to the community.

---

> > > > ### Author Response · Authors · 2024-11-25
> > > >
> > > > We greatly appreciate your constructive feedback and your elaboration on the CLEVR dataset. We are pleased to hear that the new results on GQA have successfully addressed one of your concerns.
> > > >
> > > > > Returning to Weakness One, I do not find engineering effort and computational resources acceptable justifications for avoiding additional experiments. Considering the core research question of your paper, a fair and robust evaluation is essential to substantiate your claims. I believe this is feasible, as smaller foundation models are now accessible and can be trained or fine-tuned with modest resources, including a single GPU. Incorporating such experiments would significantly strengthen the validity of your conclusions and address concerns regarding generalizability.
> > > >
> > > > Thank you for your detailed comment. We completely agree that fairness is a crucial aspect in deciding which type of representation is better for tasks like VQA. However, we would like to clarify the core question of our study: **it is not to directly compare different representation types for VQA** or to claim the superiority of any specific representation type over others. Our main goal is to analyze the relevance of OC representations in the current era of foundation models. Specifically, our goal is to evaluate how well OC representations perform in downstream visual reasoning tasks, such as VQA, and to investigate the benefits and trade-offs of OC models compared to alternative approaches, including foundation models, as baselines. In essence, we aim to answer the question: “In a visual reasoning task at hand, which model to use; an OC model or a foundation model?” by analyzing these trade-offs.
> > > >
> > > > To achieve this, we designed a large-scale empirical study that explores the effectiveness of various existing models’ representations, as outlined in the literature, without significant modifications. We also aim to fill a gap in the literature by specifically evaluating OC representations in downstream reasoning tasks, an area that has received limited attention to date.
> > > >
> > > > Therefore, while fairness is an important aspect of comparative studies, especially when making direct comparisons between different representation types, it is not central to our study and is not necessary for our findings and takeaways.
> > > >
> > > > We apologize for the confusion and hope this clarifies the core question of our work.

---

> > > > > ### Author Response · Authors · 2024-11-25
> > > > >
> > > > > > According to the findings and takeaway messages, I am sure they are correct (based on the experimental results you provided), but I agree with 7see that some (or most) conclusions are intuitive and well-known, which minimize their contribution to the community.
> > > > >
> > > > > We completely understand your point regarding the intuitive nature of some of the conclusions, and we appreciate your perspective.
> > > > >
> > > > > TL;DR: 1) We believe it is not sufficient to overlook the analysis of an idea simply because it is intuitive, and 2) In the literature, intuition hasn't always aligned with reality after proper investigation.
> > > > >
> > > > > In science, even conclusions that seem intuitive or well-known benefit from being examined carefully through a systematic approach. Many important concepts in machine learning started out as intuitive but required thorough investigation and empirical validation to fully appreciate their significance. For example, the idea of transfer learning initially seemed intuitive—that knowledge from one task could be transferred to another—but it wasn’t until systematic studies were conducted that its potential, limitations, and practical applications were fully understood.
> > > > >
> > > > > Furthermore, in the OC community, the idea of DINOSAUR — which applies the object-centric (OC) inductive bias to high-level features rather than low-level pixels— may seem intuitive at first glance. However, despite its intuitive nature, this idea has been rigorously explored and has become an influential work in the field.
> > > > >
> > > > > An additional, more nuanced example is disentanglement. Intuitively, it was believed that disentanglement is a key factor in representation learning and would generally improve generalization and performance across tasks [1]. However, subsequent research, including [2] and other follow-up studies such as [3-5] challenged this intuition and showed that disentanglement does not always result in better generalization and improved downstream task performance, and factors like hyperparameters and inductive biases on the model and data play a more significant role than previously assumed.
> > > > >
> > > > > To be more specific on one of our main findings, it is indeed intuitive to expect that, particularly on synthetic data, the OC models such as DINOSAURv2 have more explicit representations compared to the foundation model backbone alone. However, based on existing knowledge and literature in the field, it remains unclear how these representations perform on downstream reasoning tasks such as VQA and how performance trends evolve with varying downstream model capacities.
> > > > >
> > > > > Thus, in our humble opinion, intuitive ideas must still be explored and validated through careful analysis to ensure their robustness, understand the deeper implications, and provide a ground for future research.
> > > > >
> > > > > We hope this clarifies the motivation behind the analysis, and we truly appreciate your constructive comments.
> > > > >
> > > > > [1] Bengio, Yoshua, et al. "Representation Learning: A Review and New Perspectives." arXiv, 24 June 2012, doi:10.48550/arXiv.1206.5538.
> > > > >
> > > > > [2] Locatello, Francesco, et al. "Challenging Common Assumptions in the Unsupervised Learning of Disentangled Representations." arXiv, 29 Nov. 2018, doi:10.48550/arXiv.1811.12359.
> > > > >
> > > > > [3] Montero, Milton Llera, et al. "The role of disentanglement in generalisation." International Conference on Learning Representations. 2020.
> > > > >
> > > > > [4] Montero, Milton L., et al. "Lost in latent space: Disentangled models and the challenge of combinatorial generalisation." arXiv preprint arXiv:2204.02283 (2022).
> > > > >
> > > > > [5] Träuble, Frederik, et al. "On disentangled representations learned from correlated data." International conference on machine learning. PMLR, 2021.

---

> > > > > > ### Author Response · Authors · 2024-12-04
> > > > > >
> > > > > > Thank you for acknowledging our rebuttal and reflecting your positive assessment through an updated score. We sincerely appreciate the time and effort you dedicated to reviewing our paper and providing valuable feedback.

---

> > ### Comment · Reviewer_HU9k · 2024-11-25
> >
> > Thank you for the detailed rebuttal and clarifications.
> >
> > I appreciate that your study has a different focus, and I did not reject the paper based on prior results. However, I remain concerned about the use of datasets like CLEVR and other synthetic data. These datasets are not sufficiently challenging for modern methods, and this raises questions about whether your conclusions can be generalised to broader Visual Question Answering (VQA) domains. For instance, CLEVR's leaderboard (https://paperswithcode.com/sota/visual-question-answering-on-clevr) shows that research interest has significantly declined since 2021, likely because the dataset no longer provides meaningful differentiation among state-of-the-art methods. As you noted, there are already models achieving near-perfect accuracy on CLEVR, such as Neural-Symbolic VQA (99.8%) and another approach (https://arxiv.org/pdf/2104.12763v2) with 99.7% on CLEVR and 100% on CLEVR-REF+.
> >
> > The results on GQA (provided in the rebuttal phase) addressed the above concerns.
> >
> > Returning to Weakness One, I do not find engineering effort and computational resources acceptable justifications for avoiding additional experiments. Considering the core research question of your paper, a fair and robust evaluation is essential to substantiate your claims. I believe this is feasible, as smaller foundation models are now accessible and can be trained or fine-tuned with modest resources, including a single GPU. Incorporating such experiments would significantly strengthen the validity of your conclusions and address concerns regarding generalisability.

---

### Official Review · Reviewer_7see · 2024-11-03

**Soundness:** 3
**Presentation:** 4
**Contribution:** 3
**Rating:** 6
**Confidence:** 3

**Summary:**

This paper presents a comprehensive empirical study of Object-Centric (OC) representations in VQA task. The study evaluates 15 different upstream models across three synthetic datasets and one real-world dataset. The authors draw conclusions on the comparison foundation models and OC models, the relation of VQA performance and intermediate tasks or metrics, the training data efficiency and so on.

**Strengths:**

I appreciate the authors' great efforts (and also their GPU cluster's) of this empirical study. The paper shows empirical rigor through its extensive experimental design, with multiple trials across various models and datasets. The experiments brings credibility to the conclusions, while the conclusions themselves could provide valuable insights for the field.

**Weaknesses:**

- Some findings align with existing intuitions. For example, it is not suprising that large foundation models perform comparably to specialized OC models, and that their combination yields better performance. While valuable, these conclusion is intuitive and somehow obvious to the maching learning field.
- The correlation between property/attribute accuracy and overall VQA performance may be due to the characteristics or preference of the selected 4 datasets, rather than a fundamental relationship. This correlation might not generalize well to relational or reasoning or logical questions, potentially make the conclusion less convincing.
- While there are many real-world QA datasets like VCR, GQA, DAQUAR which would capture more general data intrinsics of QA question, only one real-world dataset (VQA-v2) was used in this work. The three synthetic datasets is quantitatively extensive, but might exhibit construction biases or preferences and not capture the full complexity of real-world scenarios. Focusing on more diverse real-world datasets would strengthen the conclusions

**Questions:**

Minor: I wonder how is the "640" computed on Line 315? And the "640 models" is slightly misleading, as they are actually different are experimental configurations rather than distinct model architectures.

---

> ### Author Response · Authors · 2024-11-15
>
> Thank you for your review and your positive assessment of our work. We answer your questions and concerns below:
>
> ### Weaknesses
>
> > Some findings align with existing intuitions. For example, it is not suprising that large foundation models perform comparably to specialized OC models, and that their combination yields better performance. While valuable, these conclusion is intuitive and somehow obvious to the maching learning field.
>
> Thank you for your comment. We would like to highlight the following points:
>
> 1. We appreciate the reviewer’s observation that some findings align with existing intuitions. However, we emphasize that scientific progress relies on rigorous validation, even for results that might seem obvious. While it may seem intuitive that large foundation models perform comparably to specialized OC models and that their combination further improves the performance, such outcomes have not been systematically shown on downstream reasoning tasks. Additionally, not everyone in the field shares the same intuition—some may expect the combination to perform worse and view the OC inductive bias as a potential bottleneck. By empirically testing these assumptions, we believe our work provides a strong basis for future research and offers valuable insights for the community.
> 2. We would like to highlight that despite their improvement in recent years, OC models still lag behind other types of models and foundation models in several tasks such as unsupervised segmentation in real-world settings [1]. Furthermore, the comparison between OC representations and other types of representations is still partially underexplored. The closest work that addresses this is [2] which evaluates OC representations in specific RL tasks that are significantly simpler than our VQA tasks. Therefore, it’s hard to predict how OC representations perform on harder tasks that require reasoning such as VQA without conducting a proper study to analyze it.
> 3. We would like to clarify that our focus is not solely on performance, but also on other important aspects such as the explicitness of representations and computational requirements. While performance is a key consideration, it is not the only relevant metric, and its significance depends on the specific setting and objectives at hand. Measuring other factors alongside performance is crucial for gaining a more comprehensive understanding of model behavior and evaluating their practical applicability in different contexts.
>
> > The correlation between property/attribute accuracy and overall VQA performance may be due to the characteristics or preference of the selected 4 datasets, rather than a fundamental relationship. This correlation might not generalize well to relational or reasoning or logical questions, potentially make the conclusion less convincing.
>
> We agree that there is a possibility that this correlation is due to the properties of the datasets. However, we would like to clarify that many generated questions for synthetic datasets do in fact involve relational reasoning. For example:
> * “Is the shape of the small object that is on the right side of the medium cube the same as the large thing that is behind the medium block?”
> * “How many other things are there of the same shape as the large cyan thing?”
>
> > While there are many real-world QA datasets like VCR, GQA, DAQUAR which would capture more general data intrinsics of QA question, only one real-world dataset (VQA-v2) was used in this work. The three synthetic datasets is quantitatively extensive, but might exhibit construction biases or preferences and not capture the full complexity of real-world scenarios. Focusing on more diverse real-world datasets would strengthen the conclusions
>
> We kindly refer you to our general response to all reviewers, where we explain that we are incorporating the GQA dataset as an additional real-world dataset into our study and will share the results as soon as they are ready.

---

> > ### Author Response · Authors · 2024-11-15
> >
> > ### Questions
> >
> > > Minor: I wonder how is the "640" computed on Line 315? And the "640 models" is slightly misleading, as they are actually different are experimental configurations rather than distinct model architectures.
> >
> > We apologize for the confusion. We will clarify this in the final version of the paper. This number corresponds to the total number of VQA downstream models we trained for the study. Given the number of datasets (4) and the number of different split sizes of Multi-dSprites (4; 40k, 80k, 160k, and 320k images. So 7 different datasets in total), number of models (15), number of seeds (1 for pre-trained models, 3 for other models), and the number of downstream model sizes (3; T-2, T-5, and T-15), the total number of downstream models can be calculated as:
> >
> > * For Foundation models, we will have: 6 * 7 * 1 * 3 (#foundation models * #datasets * #seeds * #downstream_models) = 126
> > * For other models on all datasets excluding VQA-v2: 9 * 6 * 3 * 3 (#models * #datasets excluding VQA-v2 * #seeds * #downstream_models) = 486
> > * And for other models on VQA-v2:  3 * 1 * 3 * 3 (#models * #datasets * #seeds * #downstream_models) = 27
> >
> > Additionally, for each dataset, we have one baseline with only the information about the questions, which will add 7 more downstream models to our study. Therefore, in total, we’ve trained 646 downstream models.
> >
> >
> > [1] Seitzer, Maximilian, et al. "Bridging the gap to real-world object-centric learning." arXiv preprint arXiv:2209.14860 (2022).
> >
> > [2] Yoon, J., Wu, Y. F., Bae, H., & Ahn, S.. “An investigation into pre-training object-centric representations for reinforcement learning.” arXiv preprint arXiv:2302.04419 (2023).
> >
> >
> > We are happy to answer any further questions you may have.

---

> > > ### Author Response · Authors · 2024-11-21
> > >
> > > Dear reviewer, thank you once again for taking the time to review our submission. We’d appreciate it if you could let us know whether the additional experiments and our responses address your concerns and if you might consider adjusting your score accordingly. We’re also open to any further feedback or questions you may have during the discussion period.

---

> > > > ### Comment · Reviewer_7see · 2024-11-25
> > > >
> > > > Thanks. I have read the revised draft and the authors' and other reviewers' comments, where our reviewers' concerns are suprisingly aligned. I really appreciate the authors' early responses, and also the additional experiments on GQA, which shows the authors‘ serious attitude on this work.
> > > >
> > > > Now, some of my concerns are well addressed. The experments on GQA make the empirical study more solid and robust. The authors also give sufficient detauils on the experiment settings. However, some major ones still exist. As also mentioned by other reviewers, some (or most) conclusions are intuitive and well-known, which minimize their contribution to the community.
> > > >
> > > > Overall, I slightly lean towards acceptance due to the solid empirical experiments and potential contributions. Since I have already given a positive rating, I would keep my score.

---

> > > > > ### Author Response · Authors · 2024-11-25
> > > > >
> > > > > Thank you for your constructive discussion and valuable feedback, which have greatly contributed to improving our work. We appreciate you acknowledging our early responses, the additional experiments on GQA, and the robustness and solidity of our empirical results and contributions. Your positive score and recognition of these aspects are greatly appreciated.

---

### Official Review · Reviewer_rPSW · 2024-11-04

**Soundness:** 2
**Presentation:** 2
**Contribution:** 2
**Rating:** 6
**Confidence:** 4

**Summary:**

This paper presents a comprehensive empirical study comparing object-centric (OC) representations with foundation models on Visual Question Answering tasks. Through extensive experiments involving 15 different upstream models, the authors demonstrate that combining object-centric bias with foundation models can achieve strong performance while reducing computational costs compared to using foundation models alone.

**Strengths:**

- Extensive empirical evaluation across different upstream and downstream models and datasets
- Experimental results thoroughly support and validate each insight presented

**Weaknesses:**

- While the study includes VQA-v2, it primarily relies on synthetic datasets. The evaluation would be more compelling if it included additional established real-world VQA datasets such as GQA or CRIC. Each image in the GQA and CRIC datasets is associated with a scene graph describing the image's objects, attributes, and relations. Moreover, questions in these datasets involve multiple reasoning skills, spatial understanding, and multi-step inference, making them ideal for analyzing object-centric models.
- The majority of claims and findings are based on synthetic datasets, which feature simpler scenes with clear object-background separation. This raises concerns about the generalizability of the findings to more complex real-world scenarios. It would be great if the author can provide additional analysis on the challenging datasets such as GQA and CRIC
- The downstream architecture is limited to BERT-style transformer encoders. The authors should explore decoder-based transformer architectures, which have become increasingly popular and achieved more favorable results than BERT-style encoders on VQA tasks in recent research.
- While the authors claim they evaluated 640 downstream models, the paper lacks sufficient detail and analysis regarding these experiments. They should provide comprehensive information about these models in the paper

**Questions:**

- How would the proposed approach perform on more challenging real-world datasets like GQA or CRIC?
- How would the findings about the benefits of object-centric representations translate to more challenging scenarios in real datasets?
- Why did the authors choose to focus on transformer encoder architectures for downstream models? Would incorporating decoder-based architectures potentially lead to different conclusions?
- Could the authors provide more detailed analysis of experiments with 640 downstream models?

---

> ### Author Response · Authors · 2024-11-15
>
> Thank you for your review of our work and the detailed feedback. We answer your questions and concerns below:
>
> ### Weaknesses and Questions
>
> > While the study includes VQA-v2, it primarily relies on synthetic datasets. The evaluation would be more compelling if it included additional established real-world VQA datasets such as GQA or CRIC. Each image in the GQA and CRIC datasets is associated with a scene graph describing the image's objects, attributes, and relations. Moreover, questions in these datasets involve multiple reasoning skills, spatial understanding, and multi-step inference, making them ideal for analyzing object-centric models.
>
> > The majority of claims and findings are based on synthetic datasets, which feature simpler scenes with clear object-background separation. This raises concerns about the generalizability of the findings to more complex real-world scenarios. It would be great if the author can provide additional analysis on the challenging datasets such as GQA and CRIC
>
> > How would the proposed approach perform on more challenging real-world datasets like GQA or CRIC?
>
> > How would the findings about the benefits of object-centric representations translate to more challenging scenarios in real datasets?
>
> We kindly refer you to our general response to all reviewers, where we explain that we are incorporating the GQA dataset as an additional real-world dataset into our study and will share the results as soon as they are ready.
>
> > The downstream architecture is limited to BERT-style transformer encoders. The authors should explore decoder-based transformer architectures, which have become increasingly popular and achieved more favorable results than BERT-style encoders on VQA tasks in recent research.
>
> > Why did the authors choose to focus on transformer encoder architectures for downstream models? Would incorporating decoder-based architectures potentially lead to different conclusions?
>
> In our work, we frame VQA as a classification task and adopt an encoder-decoder architecture with a transformer encoder to process the input and a classification head applied to the [CLS] token. A transformer encoder is well-suited for this setup because its bidirectional attention mechanism enables it to integrate and contextualize information from both the vision and language representations into the [CLS] token, which is specifically designed to represent the aggregated information required for classification tasks. However, if we were to shift to a token generation approach for the VQA task, using a transformer decoder could indeed improve the performance. On the other hand, handling VQA as a classification task helps avoid the potential influence of language variability, open-ended outputs, and other confounding factors. Despite its simplicity, this approach still aligns with our goal and effectively captures what we aim to measure: the models’ understanding of the visual scene and their relational reasoning abilities, rather than merely solving the VQA task. Moreover, it avoids the added complexity of language generation, ensuring that the results remain clearer and easier to interpret.
>
> > While the authors claim they evaluated 640 downstream models, the paper lacks sufficient detail and analysis regarding these experiments. They should provide comprehensive information about these models in the paper
>
> > Could the authors provide more detailed analysis of experiments with 640 downstream models?
>
> We apologize for the confusion. This number corresponds to the total number of VQA downstream models we trained for the study. Given the number of datasets (4) and the number of different split sizes of Multi-dSprites (4; 40k, 80k, 160k, and 320k images. So 7 different datasets in total), number of models (15), number of seeds (1 for pre-trained models, 3 for other models), and the number of downstream model sizes (3; T-2, T-5, and T-15), the total number of downstream models can be calculated as:
>
> * For Foundation models, we will have: 6 * 7 * 1 * 3 (#foundation models * #datasets * #seeds * #downstream_models) = 126
> * For other models on all datasets excluding VQA-v2: 9 * 6 * 3 * 3 (#models * #datasets excluding VQA-v2 * #seeds * #downstream_models) = 486
> * And for other models on VQA-v2:  3 * 1 * 3 * 3 (#models * #datasets * #seeds * #downstream_models) = 27
>
> Additionally, for each dataset, we have one baseline with only the information about the questions, which will add 7 more downstream models to our study. Therefore, in total, we’ve trained 646 downstream models.
>
> Regarding the request for a “more detailed analysis of experiments,” we’d be happy to provide this and will address any specific aspects you’d like clarified. As long as it does not involve new experiments with significant runtime, we should be able to include the results before the discussion deadline.
>
>
>
> We are happy to answer any further questions you may have.

---

> > ### Author Response · Authors · 2024-11-21
> >
> > Dear reviewer, thank you once again for taking the time to review our submission. We’d appreciate it if you could let us know whether the additional experiments and our responses address your concerns and if you might consider adjusting your score accordingly. We’re also open to any further feedback or questions you may have during the discussion period.

---

> > > ### Author Response · Authors · 2024-11-25
> > >
> > > Dear Reviewer,
> > >
> > > Thank you for the time and effort you have dedicated to reviewing our work. Your thoughtful feedback has been instrumental in enhancing the clarity and rigor of our paper.
> > >
> > > As the discussion period is coming to a close, we wanted to ensure that our responses along with the additional experiments on the GQA dataset have addressed your concerns. If we have successfully addressed your points, we would be grateful if you could kindly consider increasing your score. Of course, we remain happy to address any additional points if needed.
> > >
> > > Thank you once again for your valuable feedback.

---

> > ### Comment · Reviewer_rPSW · 2024-11-26
> >
> > Thank you to the authors for their responses. I really appreciate the additional experiments on GQA, and the authors' responses address most of my concerns. Therefore, I am slightly leaning towards acceptance and will increase my score.

---

> > > ### Author Response · Authors · 2024-11-26
> > >
> > > Thank you for acknowledging our responses and additional experiments on GQA. We sincerely appreciate your time and effort in reviewing our work and providing constructive feedback.

---

### Author Response · Authors · 2024-11-15
**General Response to All Reviewers**

We thank all reviewers for their helpful feedback. We are happy to see that the reviewers find our paper well-written (feeJ), with important insights for the community (7see, feeJ) supported by extensive experimentation (rPSW, 7see, HU9k, feeJ).

Common concerns were raised regarding the need for experiments on additional real-world datasets. We appreciate this valuable feedback and fully acknowledge its importance. To address this, we have started integrating the GQA dataset into our framework. Given the effort and computational resources needed, we expect to have the initial results ready a few days before the discussion deadline. We will post an update here and hope the reviewers will consider this in their assessment.

In the meantime, we will address your other concerns and questions in our responses to each review.

---

> ### Author Response · Authors · 2024-11-21
> **Additional Results on GQA**
>
> Based on the reviewers’ concerns about the need for experiments on additional real-world datasets, we have updated our submission by adding the results on the GQA dataset (highlighted in blue color). The updates in the submission are as follows:
>
> * Added three new plots for the results on GQA: two in Figure 4 and one in Figure 14. Additionally, the “Real-world Data” paragraph in Section 4.1 is updated accordingly.
>     * Figure 4 (left) shows the overall accuracies of different models w.r.t. downstream GFLOPS on GQA.
>     * Figure 4 (middle) shows the average accuracies of different models on GQA, with T-15 as the downstream model.
>     * Figure 14 (right) shows the average accuracies of different models w.r.t. downstream model sizes on GQA.
> * Added the table with the average accuracies of models on GQA, with T-15 as the downstream model (Table 14 in Appendix D).
> * Updated “Datasets” section (Section 3.3) with a brief description of GQA, and a more detailed description is added to Appendix B.
> * Added upstream and downstream training details of GQA to Appendix A.
> * Adjusted several minor parts of the main text and the appendix to include the addition of GQA.
>
> We provide a summary of the updates below:
>
> On GQA, we repeat the VQA experiments using the same pipeline on a subset of best-performing OC models and all foundation models. We use the balanced version of the dataset, with the “train” split used for training and “testdev” split used for evaluation. The new results align perfectly with our main findings:
> * **Performance Comparison**: Large foundation models, together with DINOSAURv2, outperform other models while other OC models do not perform well on GQA (see Figure 4, middle). This trend matches the results observed for VQA-v2 (see Figure 4, right).
> * **Effect of OC Bias**: We observe in Figure 14 (right) and Figure 4 (left) that DINOSAURv2 outperforms DINOv2 on T-2. On T-5, DINOv2 starts catching up and on T-15, it performs on par with DINOSAURv2. Furthermore, for all downstream model sizes, DINOSAURv2 requires significantly less downstream compute than DINOv2 (see Figure 4, left).
>
> These findings on GQA are consistent with those from other datasets, confirming our main claims (Section 4.1): Applying the OC inductive bias on top of a foundation model provides a compelling approach to get the best of both worlds. This approach achieves comparable or superior results to foundation models while significantly reducing compute requirements and providing the advantage of explicit object representations.

---

### Author Response · Authors · 2024-12-04
**Summary of Discussion Outcomes and Manuscript Improvements**

Dear AC and Reviewers,

As the rebuttal period approaches its end, we would like to offer a brief summary of the discussion outcomes and highlight the improvements made to our manuscript based on the valuable feedback we received:

* **Request for an additional real-world dataset (raised by all reviewers as a major concern)**: We added experimental results on GQA during the discussion period, which fully supported our claims in the paper. The new results and the required effort were acknowledged by all the reviewers, however only reflected in the scores of rPSW and HU9k.
* **Suggestion of using transformer decoders instead of transformer encoders as the downstream model (rPSW)**: We explained why the current pipeline is suitable and sufficient for our goal. This explanation was acknowledged by rPSW and reflected in their updated score.
* **Some of the conclusions, whether it be major or minor, are intuitive or well-known (7see and HU9k)**:
    * **7see raised concerns about the intuitive nature of conclusions, such as the combination of the OC inductive bias and a foundation model improving performance over the foundation model alone**: We emphasized that while intuitive to some, this has not been systematically tested for OC models in downstream reasoning tasks like VQA. Additionally, by analyzing performance, explicitness of representations, and computational requirements, we provide insights beyond performance alone to enhance understanding of OC representations and their applicability.
    * **Later, 7see raised a broader concern about the intuitiveness of most conclusions and maintained their score.**
    * **HU9k raised concerns about the intuitiveness of specific conclusions, such as "Consistency of the Results Across Question Types"**: We explained that while it is a minor takeaway, this conclusion has not been definitively demonstrated for OC models. Additionally, the differences in model training approaches make it plausible that certain representations might perform better on specific question types.
    * **HU9k later expressed concerns about the intuitiveness of some or most of the overall conclusions in the paper, referencing the final concern raised by 7see**: We further raised two main points: (1) it is not sufficient to overlook the analysis of an idea simply because it is intuitive, and (2) intuition does not always align with reality in the literature after proper investigation, as demonstrated e.g. in the disentanglement literature. This response was acknowledged by HU9k, as reflected in their updated score. While we were responding to HU9k, we believe this response also effectively addresses the concern raised by 7see, although this was not explicitly acknowledged.
* **Confusion about the core question of our study (HU9k)**: We clarified the core question of our study, which was accepted by HU9k, as indicated by their increased score.
* **Missing analysis of the applications of the findings (feeJ)**: We explained the main contributions of our study and the takeaway messages for the community. According to feeJ’s response, this addressed the reviewer’s concern to some extent but did not affect the final score.
* **Effect of fine-tuning the foundation model on the downstream VQA task to make comparisons more fair (feeJ)**: We described our initial experiments of fine-tuning DINOv2 on VQA and explained that while downstream performance improved slightly, the gains were not significant and introduced substantial computational costs, making comparisons with OC models less fair in terms of compute requirements. Although the reviewer acknowledged our efforts, this did not lead to an increase in the final score.

We believe we have addressed all major concerns raised by the reviewers during the rebuttal. Reviewers rPSW and HU9k notably increased their scores from 3 to 6 and from 5 to 6, respectively, recognizing the improvements made to our manuscript and expressing a positive inclination towards its acceptance. On the other hand, reviewers 7see and feeJ maintained their initial scores, without raising any further concerns or questions.

In conclusion, we are deeply grateful to the Area Chair for their guidance and support throughout the review and discussion process. We also sincerely thank the reviewers for their valuable time and constructive feedback, which have significantly improved our manuscript. We hope this summary provides helpful context regarding the efforts and progress made during the rebuttal phase.

Best regards,

The Authors

---

### Meta-Review · Area_Chair_WGch · 2024-12-25

**Metareview:**

The submission conducts an empirical study on the impact of object-centric representations for visual question answering, and aims to put the observations into perspective with "foundation models". The original submission primarily relied on synthetic datasets for their analysis, and the authors provided additional results on GQA, on which the main observations align with the original claims. After rebuttal, the submission received four borderline accept (6) ratings. All reviewers appreciate the GQA experiments as demonstration on the generalizability of the empirical study on real world benchmarks. While some of the observations might be intuitive or partially demonstrated by prior work, as pointed out by reviewer HU9k, the AC agrees with the authors' response that the systematic study presented by the submission, and the take-away messages summarized by the authors (in response to the question "I am not sure how these takeaway messages could contribute to the community") are valuable to the ICLR research community. The AC therefore agrees with the consensus reached by the reviewers and recommends acceptance of the submission to ICLR 2025.

**Additional Comments On Reviewer Discussion:**

Please find above.

---

### Decision · Program_Chairs · 2025-01-22

Accept (Poster)